**nature** COMMUNICATIONS

# Printing two-dimensional gallium phosphate out of liquid metal

Nitu Syed[1], Ali Zavabeti[1], Jian Zhen Ou[1], Md Mohiuddin[1], Naresh Pillai[1], Benjamin J. Carey[2], Bao Yue Zhang[1], Robi S. Datta[1], Azmira Jannat[1], Farjana Haque[1], Kibret A. Messalea[1], Chenglong Xu [1], Salvy P. Russo[3], Chris F. McConville[4], Torben Daeneke[1] & Kourosh Kalantar-Zadeh [1,5]

Two-dimensional piezotronics will benefit from the emergence of new crystals featuring high piezoelectric coefficients. Gallium phosphate ($GaPO_4$) is an archetypal piezoelectric material, which does not naturally crystallise in a stratified structure and hence cannot be exfoliated using conventional methods. Here, we report a low-temperature liquid metal-based two-dimensional printing and synthesis strategy to achieve this goal. We exfoliate and surface print the interfacial oxide layer of liquid gallium, followed by a vapour phase reaction. The method offers access to large-area, wide bandgap two-dimensional (2D) $GaPO_4$ nanosheets of unit cell thickness, while featuring lateral dimensions reaching centimetres. The unit cell thick nanosheets present a large effective out-of-plane piezoelectric coefficient of $7.5 \pm 0.8\,pm\,V^{-1}$. The developed printing process is also suitable for the synthesis of free standing $GaPO_4$ nanosheets. The low temperature synthesis method is compatible with a variety of electronic device fabrication procedures, providing a route for the development of future 2D piezoelectric materials.

[1] School of Engineering, RMIT University, Melbourne, VIC, 3001, Australia. [2] Institute of Physics and Center for Nanotechnology, University of Münster, Münster, 48149, Germany. [3] Chemical and Quantum Physics Group, ARC Centre of Excellence in Exciton Science, School of Science, RMIT University, Melbourne, VIC, 3001, Australia. [4] School of Science, RMIT University, Melbourne, VIC, 3001, Australia. [5] School of Chemical Engineering, University of New South Wales, Kensington, NSW, 2033, Australia. Correspondence and requests for materials should be addressed to T.D. (email: torben.daeneke@rmit.edu.au) or to K.K.-Z. (email: k.kalantar-zadeh@unsw.edu.au)

Piezoelectricity is the property of a material that allows the conversion of electrical energy into mechanical force and vice versa[1]. Recently the exploration and implementation of two-dimensional (2D) planes as piezoelectric structures has been a focus of the attention due to the promising properties of these systems[1–3]. Mechanical displacements, such as vibration, bending and stretching, are ubiquitously present in the ambient environment and 2D piezoelectric materials may facilitate their sensing and harvesting of their kinetic energy to power miniaturised devices[1,4]. The specific qualities offered by 2D materials including their lateral strength and high crystallinity along the planes, large surface area to mass ratios and compatibility with surface fabrication processes, provide the concept of 2D piezotronics with great prospect for future industries.

The observation of piezoelectricity in certain 2D materials relies on the loss of centrosymmetry that is seen for the example of doped graphene, hexagonal boron nitride and many odd-layered transition-metal dichalcogenides (TMDs) species[1,2,5,6]. Spontaneous piezoelectricity of selected other 2D materials such as transition-metal oxides (e.g., zinc oxide (ZnO))[6]; group III–V semiconductors (e.g., aluminium nitride (AlN))[6]; and metal monochalcogenides (e.g., germanium monosulfide (GeS) and tin selenide (SnSe))[7] have also been explored theoretically.

Despite significant progress and unique achievements, many critical obstacles still restrict the field of 2D piezotronics. Firstly, many 2D materials such as 2D TMDs only possess piezoelectricity for structures with an odd number of layers[6], and the strength of the piezoelectric coefficients of some of these 2D materials decreases significantly with increasing sample thickness (number of layers)[6,8]. However, this layer dependence of the piezoelectric effect may impose inconsistencies in commercial batch processes. Furthermore, in a large number of 2D systems studied to date, the piezoelectricity is confined within the in-plane piezoelectric polarisations, excluding out-of-plane operation of 2D devices[6,7]. This largely impedes the application of 2D piezoelectric materials in certain nano-electromechanical systems, where out of plane piezoelectric constant ($d_{33}$) is one of the important key factors[9]. The separation of flexoelectric and piezoelectric components of $d_{33}$ is also challenging[10]. The very recent report on the piezoelectricity of 2D ZnO provides an example of a 2D material with a large out-of-plane piezoelectric coefficient, still with lateral dimensions not exceeding several hundred microns and this does not rely on a surface synthesis processes[11]. Currently, the exploration of 2D piezoelectric materials is restricted to comparatively low temperatures, and the development of 2D materials which are able to function at elevated temperatures is still not emphasised due to the commonly observed high temperature induced instability of the piezoelectric properties of these materials. Additionally, challenges associated with achieving acceptable levels of sample homogeneity over larger areas, the synthesis of very large-area 2D piezoelectric materials at relatively low temperatures and compatibility with the current silicon processes also remain largely unsolved. These restrictions have triggered an intense quest to explore new large-area 2D piezoelectric materials beyond the currently available selection of 2D materials.

Gallium phosphate (specially α-GaPO$_4$), is a well-known piezoelectric material that is iso-structural with α-quartz[12,13]. The crystal structure of gallium phosphate (Fig. 1a, b) exhibits trigonal symmetry with cell parameters of $a = 4.87$ Å and $c = 11.05$ Å and $\gamma = 120°$[14,15]. It is considered superior to quartz for several technical applications, due to a significantly higher thermal stability, comparable quality factor ($Q$) and a higher piezoelectric coefficient[13]. Moreover, the α-phase of bulk GaPO$_4$ is stable up to 930 °C and thus the material can be very promising for high-temperature sensors, offering a nearly temperature independent

piezo effect[13,16]. Despite these unique properties, GaPO$_4$ has been essentially overlooked as a 2D piezoelectric material and the structural and electronic properties of ultra-thin GaPO$_4$ are still unknown.

GaPO$_4$ does not naturally crystallise in a stratified structure, eliminating the choice of common exfoliation techniques, and hence other processes must be developed to synthesise it[17–19]. Moreover, the deposition of high quality, defect free single crystal GaPO$_4$ is crucial, and so far, no report has addressed the growth of wafer-scale 2D GaPO$_4$ films[20]. Hence, the synthesis and deposition of large-area, high quality and homogeneous ultra-thin GaPO$_4$ nanosheets will have a major impact on 2D piezotronics.

In this work, we have succeeded to synthesise extraordinary large 2D GaPO$_4$ sheets on suitable substrates using a liquid metal-based synthesis process. Recently, liquid elemental gallium and their alloys have drawn significant attention for its use in the printing deposition of 2D materials[21]. The reported method takes advantage of the self-limiting atomically thin oxide layer that naturally grows on the surface of liquid gallium, which can be transferred onto a substrate using our devised van der Waals printing technique. The 2D GaPO$_4$ synthesis process described here relies on harvesting the gallium oxide (Ga$_2$O$_3$) skin, followed by a chemical vapour reaction process using phosphoric acid at 350 °C. The phosphatisation process is conducted at relatively low temperature, which is compatible with existing industrial processes. We investigate the vertical piezoelectricity for one to several unit cell thick 2D GaPO$_4$ sheets, using a combination of piezoresponse force microscopy (PFM) and density functional theory (DFT) simulations. The experimentally measured vertical piezoelectric coefficient for unit cell thick GaPO$_4$ is found to be $7.5 \pm 0.8$ pm V$^{-1}$, in good agreement with the DFT calculations. The vertical piezoelectric behaviour of 2D GaPO$_4$ nanosheets with different thicknesses is also demonstrated in this work. Thus, we report excellent out of plane piezoelectric performance of the synthesised 2D GaPO$_4$. Additionally, the developed process is also utilised to fabricate free-standing 2D GaPO$_4$ membranes over micro-fabricated square holes.

## Results

**Printing and synthesis of 2D GaPO$_4$.** The synthesis of ultra-thin GaPO$_4$ is carried out in a two-step process consisting of van der Waals exfoliation followed by a chemical vapour phosphatisation step. A schematic illustration of the van der Waals synthesis technique for printing 2D Ga$_2$O$_3$ sheets is depicted in Fig. 1c. When a pristine liquid gallium droplet is exposed to ambient atmospheric conditions, a mechanically robust, atomically thin gallium oxide layer grows on its surface in a self-limiting reaction. Synchrotron-based studies of the liquid gallium interface have revealed that the electron density profile features a prominent minimum of the electron density distribution at the boundary between the liquid metal, and its naturally grown surface oxide[22]. This finding indicates that there are no covalent bonds between the oxide layer and the parent metal. Furthermore, liquid gallium metal is a monatomic liquid which is by default non-polar, limiting the possibility of interaction further. Hence, interaction between the liquid metal and the surface oxide are expected to be minimal. The absence of a solid crystal structure impedes cumulative atomic interactions of liquid metal over large areas rendering any weak interactions that may occur to be localised, inhibiting macroscopic attachment[23]. The absence of covalent bonds between the liquid metal together with the liquid state of the parent metal hence lead to minimal interactions between the oxide and the liquid metal. The van der Waals interactions between the surface oxide and the transfer substrate, on the other hand, comprise of more robust forces between permanent

dipoles. The presence of a crystalline lattice within the oxide as well as the substrate ensure that interactions may occur over larger areas, leading to macroscopic attachment[24,25]. The high surface tension of liquid gallium further ensures that the vast

majority of the liquid metal separates from the oxide during the exfoliation process. Thus minimal metal inclusions are found in the large exfoliated nanosheets, which are furthermore consistent with our previous reports for other metals[26]. As such, the interfacial oxide skin can be effectively transferred to a $SiO_2$/Si substrate that is brought into the close contact with the liquid gallium surface (Fig. 1c).

The schematic setup of the chemical vapour phase reaction system to transform the 2D $Ga_2O_3$ sheets into 2D $GaPO_4$ is elucidated in Fig. 1d. The phosphatisation of $Ga_2O_3$ is conducted at a relatively low temperature (300 °C–350 °C) that is compatible with the existing electronic device fabrication processes. The method for growing large-area $GaPO_4$ sheets is further found to be highly reproducible as the process was repeated in excess of 50 times and always led to uniform, continuous, laterally large atomically thin $GaPO_4$ films (presented in Supplementary Fig. 3a, b) with reproducible properties. Furthermore, it is observed that the synthesised 2D $GaPO_4$ is stable up to 600 °C (more details regarding the thermal stability of the synthesised $GaPO_4$ flakes are presented in Supplementary Note 3 and Supplementary Figs. 14, 15). Degradation of the $GaPO_4$ nanosheets is found to begin when the samples are annealed at 700 °C.

**Characterisations of 2D $GaPO_4$.** Atomic force microscopy (AFM) investigation is carried out to determine the surface morphology and thickness of the synthesised 2D films. Due to the limited scan area size of AFM, only a small section of this ultra large $GaPO_4$ sheet is presented in Fig. 2a. The typical step-height from the substrate to the $GaPO_4$ nanosheet is found to be approximately 1.1 nm for the majority of the synthesised 2D crystals, which is in excellent agreement with the unit cell dimension of trigonal $GaPO_4$ in the c-direction ($c = 11.05$ Å)[15]. Occasionally some $GaPO_4$ nanosheets of differing thicknesses are also observed during the synthesis process (details are provided in Supplementary Note 2 and Supplementary Fig. 5). Localised inclusions of metallic gallium are only occasionally encountered along the edges of the extraordinarily large sheet. Imaging of an as-grown $GaPO_4$ nanosheet (on $SiO_2$/Si substrate) using optical microscopy (Supplementary Fig. 3) revealed lateral dimension reaching several millimetres, approaching centimetres. The developed exfoliation technique is confirmed to reproducibly result in uniform, centimetre-scale $GaPO_4$ nanosheets in repeated tests (inset of Supplementary Fig. 3b).

Transmission electron microscopy (TEM) is further used for revealing the structural features and crystal structures of the

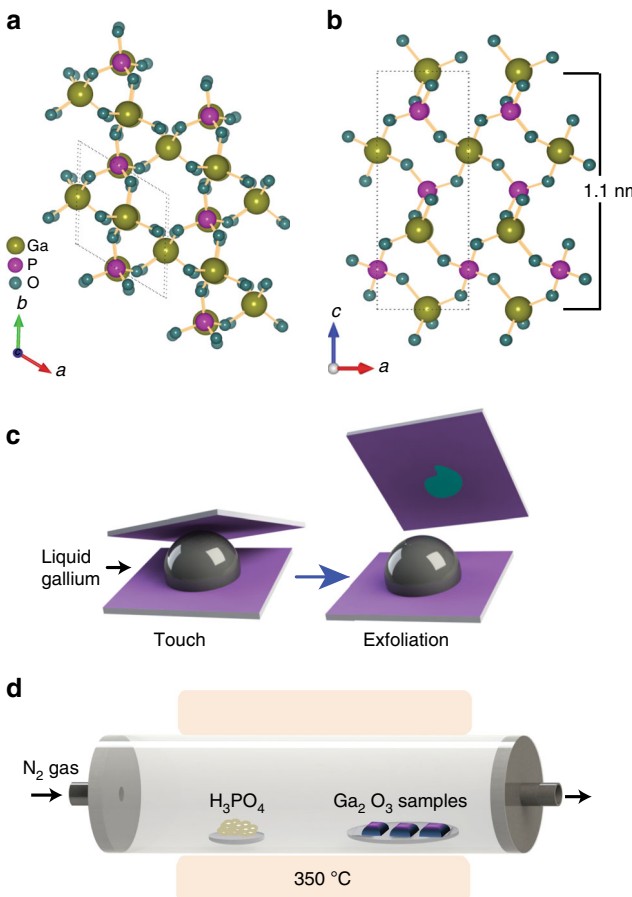

**Fig. 1** Crystal structure and printing process of 2D $GaPO_4$ nanosheets. Ball and stick representation of the synthesised $GaPO_4$ crystal: **a** top view and **b** side view showing an out-of-plane structure exhibiting unit cell parameter $c = 11.05$ Å. **c** Schematic illustration of the van der Waals 2D printing technique of $Ga_2O_3$ nanosheet from liquid metal gallium. **d** Schematic setup for the chemical vapour phase reaction system used for synthesising $GaPO_4$ nanosheets

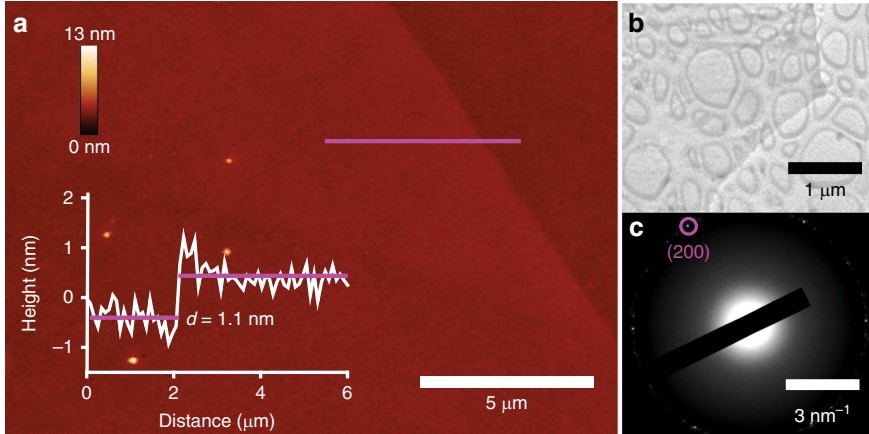

**Fig. 2** Morphology and TEM characterisations of the printed 2D $GaPO_4$ film. **a** AFM topography of a $GaPO_4$ nanosheet and height profile along the magenta line. **b** TEM micrograph of the $GaPO_4$ film. **c** The SAED pattern of the TEM micrograph

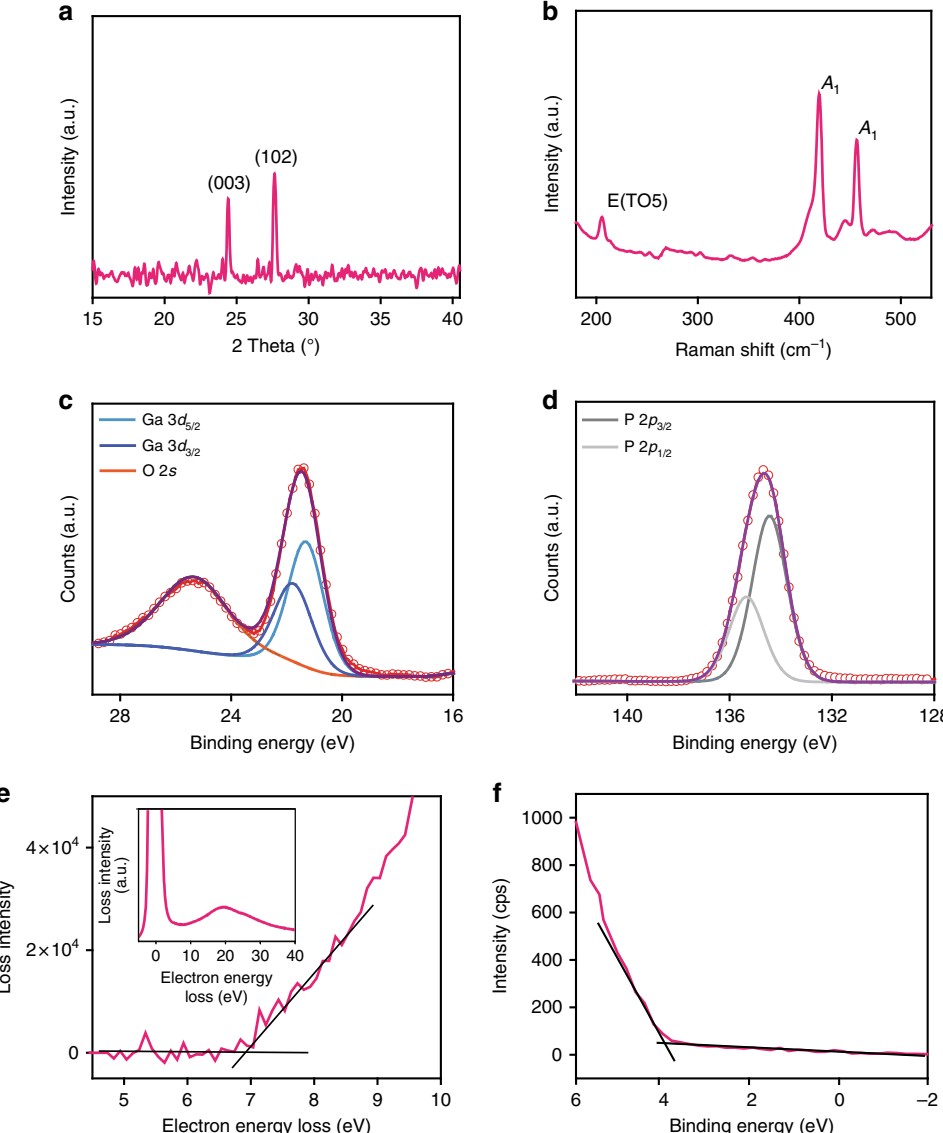

**Fig. 3** Material characterisations and electronic band properties of 2D GaPO$_4$. **a** XRD and **b** Raman spectra of the synthesised GaPO$_4$ (thick nanosheets). XPS results of **c** Ga 3d and **d** phosphorous 2p regions of the synthesised GaPO$_4$. **e** Enlarged view of EELS for the estimation of the fundamental bandgap and the extended EELS spectrum (inset). **f** XPS valence band analysis of GaPO$_4$

synthesised nanosheets. TEM samples are produced by utilising a TEM grid as a substrate, rather than the SiO$_2$/Si wafer. The TEM micrograph in Fig. 2b exhibits the translucent sheet-like morphology of the synthesised 2D GaPO$_4$. The crystalline structure of the synthesised GaPO$_4$ is confirmed by the corresponding selected area electron diffraction (SAED) pattern displayed in Fig. 2c. The SAED pattern taken on the isolated GaPO$_4$ nanosheets, indicates the observed lattice spacing is 0.21 nm corresponding to the d-spacing value of (200) plane of trigonal GaPO$_4$. Overall, the TEM results support and substantiate the morphological conclusions drawn from the AFM observations.

X-ray diffraction (XRD) measurement is employed to investigate the crystal structure of the synthesised GaPO$_4$. The XRD pattern of the as-deposited flake (Fig. 3a) reveals two dominant peaks at 24.4° and 27.3°, corresponding to the (003) and (102) planes of trigonal α-GaPO$_4$, respectively[14,17,27].

Raman spectroscopy is utilised to further validate the composition of the GaPO$_4$ sheets synthesised on a glass substrate (Fig. 3b). The Raman peaks at ~420.5 and ~456.6 cm$^{-1}$ can be

assigned to vibrations involving internal bending of the PO$_4$ tetrahedra, which can be denoted as A$_1$ vibration modes[28]. On the other hand, the 300 cm$^{-1}$ Raman peak that exists in the bulk system, which is considered to be a bending mode along the z-axis[28,29], is absent in the 2D sample due to geometrical confinement. The Raman peak at 206.2 cm$^{-1}$ can be ascribed to the decoupled bending mode (E-TO5) of the PO$_4$ tetrahedra along the x–y-axis[29]. Compared with the bulk GaPO$_4$, this peak is more prominent due to the geometrical confinement along z-axis in the 2D sample. Ga$_2$O$_3$ also features peaks at ~198 and ~415 cm$^{-1}$, which are in close proximity to the characteristic Raman modes of GaPO$_4$[30], however, several Raman features of Ga$_2$O$_3$ (i.e., peaks at ~167, ~320, ~344 and ~475 cm$^{-1}$) are absent in our spectrum, confirming that GaPO$_4$ is successfully synthesised and Ga$_2$O$_3$ is quantitatively converted during the vapour phase reaction. Another interesting observation is that the Raman peak intensity for 456.6 cm$^{-1}$ mode is significantly reduced compared to that of 420.5 cm$^{-1}$ for 2D GaPO$_4$ in comparison to the bulk counterpart[28,29]. It has been reported that the A$_1$ Raman mode is sensitive to the free charge carrier density in graphene and 2D

metal chalcogenides[31–34]. 2D GaPO$_4$ has a wider bandgap which is likely to include more trap states. The emergence of more polar 2D GaPO$_4$ modifies the interaction between phonon and free charge carriers, generated by the trap states, within the 2D material, leading to phonon self-energy renormalisation. Consequently, phonons are weakened, causing the intensity of charge-sensitive $A_1$ Raman mode to be reduced[31–34]. Therefore, the observation from Raman spectra suggests that there is an increase in the dipole intensity in 2D GaPO$_4$.

X-ray photoelectron spectroscopy (XPS) is employed to obtain the chemical bonding states of the synthesised 2D GaPO$_4$ material. Fig. 3c, d shows the spectra of Ga 3$d$ and phosphorous 2$p$ regions, respectively for the synthesised GaPO$_4$ nanosheet. The characteristic main broad peak at 21.4 eV in the Ga 3$d$ region, is fitted to identify the 3$d_{5/2}$ and 3$d_{3/2}$ components at ~21.3 and ~21.7 eV, respectively, signifying the Ga$^{3+}$ state in GaPO$_4$. The characteristic gallium peak for Ga$_2$O$_3$ located at ~20.4 eV is not observed, demonstrating quantitative transformation of GaPO$_4$ (Supplementary Fig. 4c). Additionally, no metallic gallium peak (Ga$^0$) is detected in the Ga 3$d$ region. The peak centred at 25.2 eV incorporates the broad O 2$s$ feature[26]. The main broad phosphorous 2$p$ peak centred at ~134.6 eV corresponds to the doublets P2$p_{3/2}$ and P2$p_{1/2}$, which is in agreement with the expected phosphorous 2$p$ region present in GaPO$_4$. The observed binding energies are all consistent with the previously reported values for GaPO$_4$[35–37].

These XPS data, and the all other above-mentioned characterisation techniques, support the conclusion that all of the deposited 2D GaPO$_4$ nanosheets show excellent consistency. The characterisation of the GaPO$_4$ is further corroborated by the attenuated total reflectance Fourier transform infra-red spectroscopy (FTIR) that is presented in Supplementary Fig. 4a.

Figure 3e represents the enlarged view of low loss electron energy loss spectroscopy (EELS). spectrum of GaPO$_4$ for the determination of electronic bandgap. The bandgap value of GaPO$_4$ is assessed by extrapolating the linear fit of the loss intensity to electron loss energy as illustrated in Fig. 3e. Therefore, the estimated fundamental bandgap value is approximately 6.9 eV, which shows an excellent agreement with previous reports[38–41], confirming 2D GaPO$_4$ to be a wide bandgap material. The inset (Fig. 3e) shows the extended EELS where the highest intense peak signifies the zero loss peak[42]. XPS valence band analysis is conducted to further explore the electronic properties of GaPO$_4$. The XPS valence band spectrum (Fig. 3f) represents the energy difference of 3.80 eV between the valence band maximum and Fermi level, indicating near-intrinsic behaviour of the 2D GaPO$_4$.

**Out-of-plane piezoelectric properties of 2D GaPO$_4$.** The non-centrosymmetric nature of unit cell thick GaPO$_4$ across the out-of-plane direction is evident from Fig. 1b, inducing a non-zero vertical piezoelectric response. The fundamental piezoelectric characteristics of the synthesised GaPO$_4$ thin film are explored by PFM, the most widely and extensively used technique to characterise nanoscale piezoelectric phenomena[5,43–48]. During the herein presented PFM measurements, the tip and surface electrostatic interaction is ignored, since the tip-field interaction is significantly reduced by using conductive tips with a higher force constant (3 N/m)[48]. The vertical piezoelectric response of unit cell thick GaPO$_4$ nanosheets ($d = $~1.1 nm, see Fig. 4a) is measured by applying an electrical field normal to the surface of the flake through a conductive AFM tip. The thicker areas at the edges of the sheets can be attributed to folding and restacking of the monolayer along the flake boundaries during the van der Waals exfoliation. Fig. 4b–f illustrate the vertical

piezoresponse amplitude profiles for this ultra-thin crystal under different drive voltages ranging from 0 to 4 V. The corresponding PFM phase images for different excitation voltages are also presented in Supplementary Fig. 7. No PFM amplitude and phase difference is observed between GaPO$_4$ and the SiO$_2$/Si substrate (a natural reference to the PFM signal) at 0 V. However, a non-zero vertical piezoresponse signal is observed at the thick edges of the GaPO$_4$ nanosheet even at 0 V, which is probably due to the deviation of the perfect flatness and non-uniform thickness along the boundaries, resulting in a possible non-piezoelectric polarisation component[5].

The piezoresponse displacement is found to increase steadily with higher driving bias. The insets represent the statistical distributions of piezoresponse amplitude variations of the GaPO$_4$ sheet and the substrate. The statistical distributions of the piezoresponse amplitude variations provide the opportunity to characterise the overall effective piezoelectric constant ($d^{\text{eff}}_{33}$) for the entire flake area[11]. From the piezoresponse amplitude, we characterise $d^{\text{eff}}_{33}$ of GaPO$_4$ sheets quantitatively[5,44–46] (details concerning the $d^{\text{eff}}_{33}$ calculation are provided in the method section). The piezoelectric displacement as a function of applied AC voltage is plotted in order to quantify the piezoelectric coefficient of a unit cell thick GaPO$_4$ nanosheet. The piezoresponse displacement shows a linear relationship with the driving voltage (Fig. 4g) providing a value of $d^{\text{eff}}_{33}$ of approximately 7.5 ± 0.8 pm V$^{-1}$ from the linear fit[44,46], which is approximately two times larger than that of bulk GaPO$_4$ crystal[49]. The improvement in the value of $d^{\text{eff}}_{33}$ can be possibly due to a slightly disordered crystal structure exhibited by the 2D GaPO$_4$ films. From the Raman spectra (Fig. 3b) it is observed that there are significant shifts to lower wavenumbers for both of the ~420.5 and ~456.6 cm$^{-1}$ peaks in 2D GaPO$_4$ with reference to their bulk values, indicating the weakening of optical phonon vibration modes ($A_1$) of the PO$_4$ tetrahedra along $z$-direction[28,29]. As a result, the PO$_4$ tetrahedra in the 2D GaPO$_4$ crystal structure has relatively low stability and trends to become more disordered along the $z$-direction when external stimuli are applied, due to the crystal's enhanced asymmetry compared to the bulk counterpart. Therefore, the piezoelectric response resulted in the $z$-direction is expected to augment in the 2D form.

The vertical piezoresponse is also explored for GaPO$_4$ nanosheets of different thicknesses (Supplementary Fig. 9). The obtained experimental piezoelectric constant is measured to be within the range from 6.9 to 7.2 pm V$^{-1}$ for two to four unit cell thick (~2.2–4.3 nm) GaPO$_4$ sheets (Supplementary Fig. 9a–c, e–g). However, more gradual thickness dependence behaviour of $d^{\text{eff}}_{33}$ is observed for thicker GaPO$_4$ films which are prepared from multiple van der Waals printing process as discussed in Supplementary Note 2. Two PFM measurements for the thicknesses of 8 and 11 nm films resulted in the $d^{\text{eff}}_{33}$ values of 6.6 and 6.1 pm V$^{-1}$, respectively, demonstrating a possible decrease towards that of the bulk value reported in previous literature[49].

DFT calculations are performed to ensure the veracity and the consistency of our experimental measurements. To facilitate the calculation of unit cell polarisation, a periodic 2 × 1 supercell of GaPO$_4$ with a (100) plane is cut from the trigonal bulk phase and the top and bottom oxygen atoms in the slab are stabilised (Supplementary Fig. 2b). Single and two unit cell thick slabs are constructed from the 2 × 1 supercell. The thickness of the one unit cell thick slab is ~1.1 nm, corresponding to the experimentally determined sheet thickness, and ~2.2 nm for the two unit cell thick sheet. To calculate the vertical piezoelectric constant $d_{33}$, an external electric field, $E$ is applied to the slab perpendicular to the surface, and next the strain of the slab in the direction

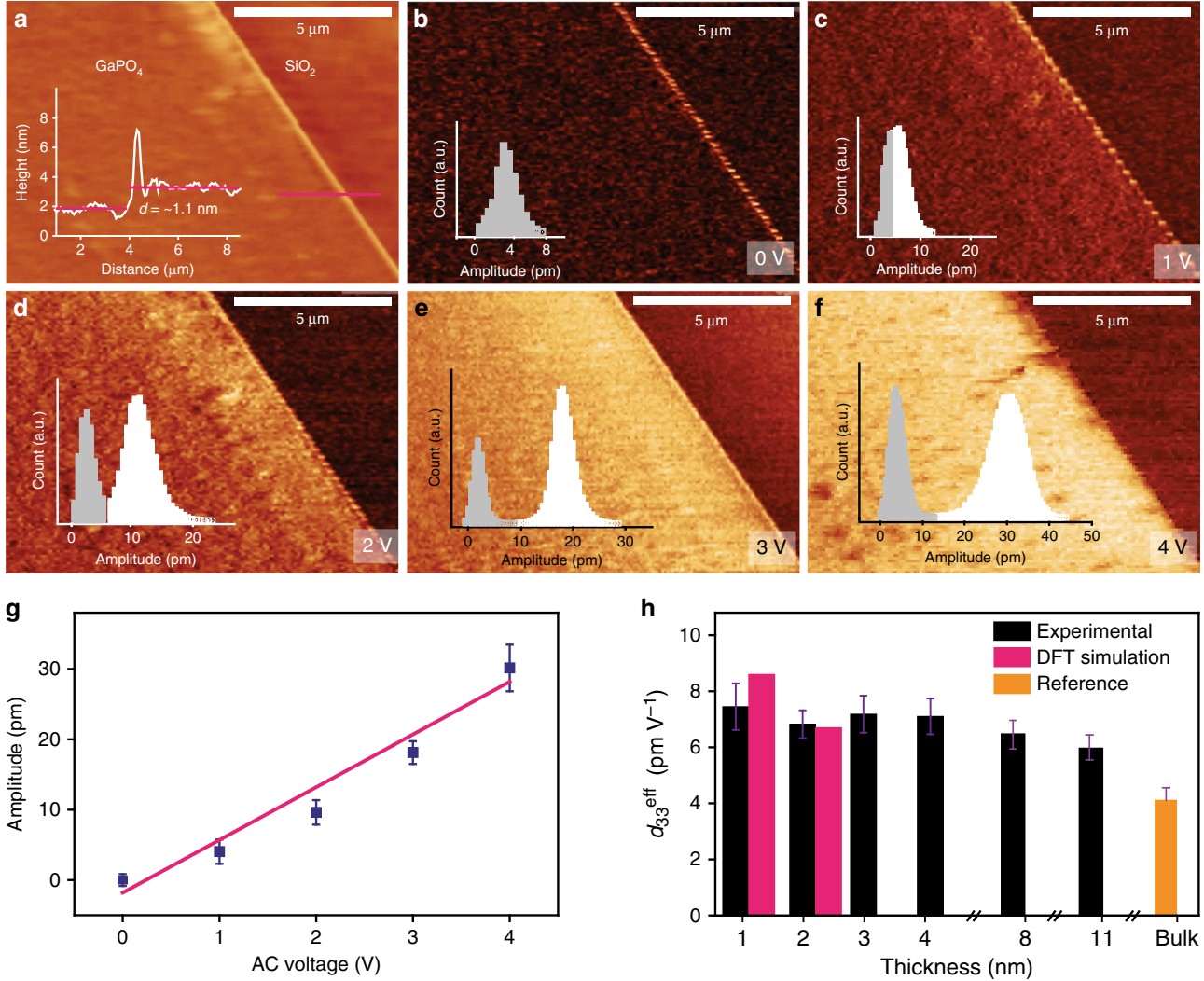

**Fig. 4** Characterisations of out of plane piezoelectricity of 2D GaPO$_4$. **a** AFM topography of a unit cell thick GaPO$_4$ nanosheet. **b-f** Vertical piezoresponse amplitude profiles at different AC driving voltages. The insets represent the statistical distribution of the piezoresponse amplitude variations of GaPO$_4$ film (white colour) and the substrate (grey colour). **g** Average piezoresponse amplitude as a function of the applied AC voltage extracted from the statistical distributions of the amplitude variations of GaPO$_4$ and the substrate. Error bars signify the standard deviations which are introduced to indicate uncertainty of the measurements. **h** Value of $d^{eff}_{33}$ for GaPO$_4$ films with different thicknesses. The bulk value for $d_{33}$ is extracted from ref.[49]. Error bars signify the deviations of slope of piezoelectric amplitude for the driving bias voltage (AC) for experimental $d^{eff}_{33}$. DFT simulations of 2D GaPO$_4$ nanosheets of higher thicknesses are not reported due to difficulties with accuracy and energy convergence

perpendicular to the surface is calculated. The strain under an applied electric field is calculated using Eq. (1):

$$\eta(E) = \frac{(L(E) - L_0)}{L_0} \quad (1)$$

where $\eta$ is the strain, $L(E)$ is the equilibrium length of the slab along the $z$-axis (perpendicular to the surface direction) under the applied $E$ field and $L_0$ is the equilibrium length under zero applied field. The magnitude of the piezoelectric constant can be defined as the variation in the strain $\eta_j$ in terms of the variation in the applied electric field $E_i$ at zero stress ($\tau$)[50] as:

$$d_{ij} = -\left(\frac{\partial \eta_j}{\partial E_i}\right)_{\tau=0} \quad (2)$$

In the case of $i = j = 3$ (in the Voigt notation), $d_{33}$ refers to the strain and the applied $E$ field in the $z$-direction, perpendicular to the surface. We interpret the minus sign in Eq. (2) to mean

that the piezoelectric constant is negative if the slab undergoes expansion (positive strain) along $z$-axis as a response to an increasing external field.

For the unit cell thick slab an applied electric field of $2.6 \times 10^7$ V cm$^{-1}$ is used (corresponds to an applied voltage of ~2.6 V), while a slight higher field strength of $3 \times 10^7$ V cm$^{-1}$ is used for the bilayer slab. We calculated a theoretical value of the piezoelectric constant, $d_{33}$ to be approximately 8.5 and 6.7 pm V$^{-1}$ in magnitude, respectively, for the single and two unit cell thick slabs indicating the slab undergoing positive strain in response to an increasing field, which is consistent with experiments. The value of experimental $d^{eff}_{33}$ for the single unit cell thick GaPO$_4$ sheet is comparable with DFT calculations with a slight mismatch. On the other hand, the experimental value of $d^{eff}_{33}$ (~6.8 ± 0.5 pm V$^{-1}$) for two unit cell thick GaPO$_4$ sheet is in good agreement with the DFT computational results. This demonstrates that there is indeed out of plane piezoelectric property for the thicker (two unit cell thick) GaPO$_4$ sheets.

The discrepancy between the real piezoelectric performance and the simulation result is possibly due to the difference between the real 2D sheet and its assumed model (either physical or mathematical). During our DFT simulation the developed unit cell thick GaPO$_4$ is considered to be a perfectly flat sheet without any defect. Additionally, the DFT calculations are performed considering a completely homogeneous applied electric field normal to the surface of the entire flake area, which is in obvious contrast with the real piezoelectric measurement of the nanosheets[48].

Finally, a comparative assessment between the out of plane piezoelectric coefficient of 2D GaPO$_4$ and some other well-established bulk and 2D piezoelectric materials is provided in the Supplementary Information (Supplementary Tables 1 and 2). It is noteworthy that the out of plane piezoelectric coefficient of our synthesised 2D GaPO$_4$ outperforms many of the previous reported results[5,51,52].

**Piezoelectric and elastic properties of free-standing GaPO$_4$.** Free-standing GaPO$_4$ is further used for exploring possible substrate effects, assessing the effect of flexoelectric component and also for obtaining the elastic modulus of the film. The elastic modulus is of particular importance since it allows assessing the suitability of the developed films for various applications. The surface topography of a free-standing GaPO$_4$ is depicted in Fig. 5a. Additional AFM topography and scanning electron microscopy (SEM) images of these free-standing sheets are also presented in Supplementary Figs. 10, 11. It is evident from the morphological study that the surface of the free-standing GaPO$_4$ bulges upward with respect to the substrates surface. The substrate areas surrounding the cavity edges are expected to firmly hold the large and continuous 2D gallium oxide sheet. The difference in van der Waals forces over and surrounding the cavities causes the gallium oxide skin to bulge upward when the substrate is lifted off the liquid metal during synthesis. The bulging feature of the free-standing flake may also be enhanced by the air pocket that is trapped inside the cavity[53]. The wrinkles on the surface of the free-standing and bulged GaPO$_4$ (Supplementary Fig. 10f), which are not prominently visible in free-standing Ga$_2$O$_3$, are assumed to be caused by phosphatisation process. A likely explanation for this effect is the expected volume change caused by the insertion of phosphorous and additional oxygen atoms into the film during the reaction.

Figure 5b–d elucidate the vertical piezoresponse under different excitation voltages for a free-standing GaPO$_4$. Relatively high piezoresponse deflection signals around the cavity edges are observed (even at 0 V). The possible origin of the non-zero signal near the cavity edges is the emergence of additional strain gradients that arise due to the sharp depth profile of the cavity[5]. Hence, we focus on the flat and uniform surface area of free-standing GaPO$_4$ for further analysis (area A, Fig. 5a, Supplementary Fig. 12) to avoid any contribution of non-zero out-of-plane polarisation. No significant amplitude contrast is evident at high voltages between the free-standing GaPO$_4$ (area A) and the supported film on the substrate (Fig. 5c, d), indicating a probability of having nearly identical $d^{eff}_{33}$ coefficients for the free-standing and supported films (details of the piezoresponse amplitude variations for the free standing and supported GaPO$_4$ film is provided in Supplementary Fig. 12). This demonstrates that the flexoelectric component is negligible and that the majority of the out-of-plane displacement is due to the piezoelectricity of 2D GaPO$_4$.

The nano-mechanical property of free-standing GaPO$_4$ is mapped using peak-force quantitative nano-mechanical (PF-QNM) mapping in order to explore the mechanical strength

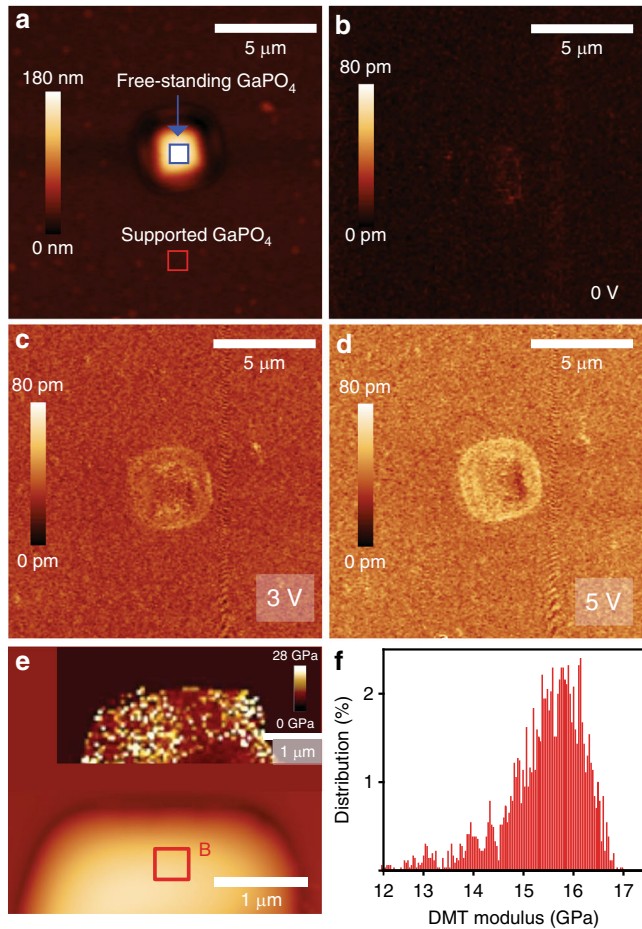

**Fig. 5** Piezoelectric and elastic properties of free-standing GaPO$_4$ nanosheet. **a** AFM topography and **b–d** vertical piezoresponse of free-standing GaPO$_4$ nanosheet at different AC drive excitation. **e** AFM image of a free-standing GaPO$_4$ flake with DMT modulus map (inset). **f** DMT modulus distributions measured for the flat section (area B of **e**) of the GaPO$_4$ sheet. The Gaussian mean value of the modulus is found around 15.66 ± 0.0095 GPa

of the synthesised GaPO$_4$. This method has been previously been applied successfully to other examples of thin films[54,55]. Quantitative results during the PF-QNM scan can be presented as Derjaguin–Mueller–Toporov (DMT) modulus distributions (stiffness distribution). The topographical image of a free-standing GaPO$_4$ sheet during PF-QNM scan is illustrated in Fig. 5e along with the DMT (elastic) modulus map (inset). The DMT modulus distribution presented in Fig. 5f reveals that the modulus situates mainly in the range of 12–16.5 GPa. This large elastic modulus demonstrates that the synthesised 2D GaPO$_4$ sheets are suitable for the manufacture of free-standing membrane based piezoelectric applications. The peak-force quantitative nano-mechanical mapping is next used for estimating the breakdown threshold load for the free-standing GaPO$_4$ flake by varying the force applied to the tip. The threshold force is found at around 200 nN to break the GaPO$_4$ membrane (Supplementary Fig. 13).

## Discussion
Liquid metals present many potential applications[56–63], and this work signifies one of the major capabilities for this family of materials, that is their implementation as a reaction media for

printing 2D materials which can be used in a variety of applications.

The work introduces a scalable and facile method for synthesising a previously inaccessible, ultra-thin 2D material, $GaPO_4$, by harvesting the naturally occurring oxide skin of liquid gallium followed by a vapour phase phosphatisation process. The unique synthetic approach is simple, cost effective and utilises relatively low temperatures, providing new avenues for creating extraordinary large-area, highly uniform 2D $GaPO_4$ films. Furthermore, these films are compatible with high-temperature fabrication procedures used for surface processed chips and high-temperature applications up to 600 °C. Here we demonstrate the first experimental evidence of strong out-of-plane piezoelectricity of a unit cell thick $GaPO_4$ sheet which is further confirmed by the DFT simulations. The observed piezoelectricity for unit cell thick $GaPO_4$ nanosheet is relatively high, achieving experimental and theoretical $d_{33}$ of 7.5 ± 0.8 and ~8.5 pm V$^{-1}$, respectively, which holds promise for 2D material based piezotronic sensing and energy harvesting. We also report the vertical piezoelectric response for free-standing $GaPO_4$, which is found to be similar to the supported films. The findings highlight that ultra-thin $GaPO_4$, can be a promising candidate for nano-electromechanical systems requiring high-temperature resistant piezoelectric materials. It can further be exploited for creating surface mounted piezo-responsive sensing elements. Overall, this study leads to promising frontiers towards embedding atomically thin piezoelectric building blocks into large-area devices, which may be utilised to harvest the kinetic energy of various mechanical vibrations to power flexible devices.

## Methods

**Materials**. Gallium (Ga, 99.99%) was purchased from Roto Metals. Phosphoric acid (crystalline $H_3PO_4$; 99.98%) was supplied from Sigma-Aldrich. All chemicals were used as received.

**Two-step liquid metal-based synthesis of 2D $GaPO_4$ sheets**. The synthesis of 2D $GaPO_4$ nanosheets involves a two-step process. Ultra-thin $Ga_2O_3$ sheets were first directly exfoliated onto a substrate from liquid elemental gallium. The isolated film was subsequently phosphatised using a chemical vapour phase reaction between gallium oxide sheets and phosphoric acid.

**Van der Waals printing process of 2D $Ga_2O_3$ sheets**. 2D $Ga_2O_3$ nanosheets were printed from liquid gallium following the liquid metal van der Waals exfoliation method reported in our previous work[26]. Gallium melts at 29.76 °C therefore; the process was conducted on a hotplate at 50 °C. A self-limiting atomically thin gallium oxide skin forms on the metal surface of gallium droplets, when they are exposed to ambient air[21]. The freshly formed gallium oxide skin is exfoliated by touching the liquid metal surface with a suitable substrate (SiO$_2$ wafer, glass and TEM grids) typically with the aid of tweezers. A gallium droplet was placed on a SiO$_2$ (300 nm)/Si substrate (Fig. 1c). After touching the gallium oxide skin, the substrate was separated very carefully. Using this technique atomically thin gallium oxide sheets with large lateral dimensions exceeding several millimetres and reaching centimetres can be exfoliated onto the surface of the substrate. The dimensions of the 2D oxide sheets varied with the diameter of the droplet (more details regarding the printing process are provided in Supplementary Note 1 and Supplementary Fig. 1). Thicker layers of $GaPO_4$ were obtained by repeating the van der Waals printing process as presented in Supplementary Note 2.

**Transformation of 2D $Ga_2O_3$ to 2D $GaPO_4$ sheets by phosphatisation**. The synthesised ultra-thin gallium oxide sheets were phosphatised by means of a chemical vapour method in a horizontal quartz tube furnace, which utilises the physical transport of the source vapour to the target (Fig. 1d). For the phosphatisation technique, phosphoric acid ($H_3PO_4$) powder was placed on an alumina boat as source material and heated to 350 °C at a rate of 15 °C/min to induce evaporation. The $Ga_2O_3$ samples were placed upside down on another alumina boat located in a comparably low temperature region (around 300 °C) of the tube furnace. Nitrogen gas with a flow rate of 0.6 L/min was used for transporting vapours of $H_3PO_4$ towards the $Ga_2O_3$ samples. As the $H_3PO_4$ vapour reaches the low temperature region, it starts to transform the thin $Ga_2O_3$ layer into $GaPO_4$. The optimal duration for the entire process was found to be 75 min after the temperature of the system was saturated to 350 °C. Placing samples in the upside down position ensures that there is no accumulation of phosphoric acid on the

synthesised nanosheets. The furnace was allowed to cool naturally to room temperature when the synthesis was completed. Any remaining residue of $H_3PO_4$ was finally removed from the synthesised 2D $GaPO_4$ samples by blowing $N_2$ gas. A nitrogen glove-box or desiccator was later used for prolonged storage of the samples.

**Exfoliation of free-standing 2D $GaPO_4$ sheets**. Free-standing $GaPO_4$ sheets were synthesised on a SiO$_2$/Si wafer with arrays of square microcavities as shown in Supplementary Fig.10b. The fabrication of the cavities was conducted by oxidising a p-doped silicon wafer with a 300-nm thick SiO$_2$ layer. Square cavities with approximately 2–3 μm length were first patterned and next focused ion beam (dual beam FIB/SEM) was utilised to etch the squares to a depth of 1–2 μm, leaving a series of holes on the wafer. Finally, 2D $GaPO_4$ sheets over the square cavities were directly synthesised following the liquid metal van der Waals exfoliation techniques and phosphatisation process as previously discussed.

**Piezoresponse force microscopy**. PFM measurements were carried out using a Bruker AFM (Bruker Dimension Icon). NanoScope® 1.8 software was used for data acquisition and analysis. Two types of Pt/Ir coated conductive AFM tip SCM PIT V2 (resonance frequency 75 kHz, spring constant 3 N/m, radius ~25 nm) and SCM-PIC-V2 (resonance frequency 10 kHz, spring constant 0.2 N/m, radius ~25 nm) were used to obtain the PFM results for the supported and free-standing nanosheets, respectively. The schematic representation of PFM setup is provided in Supplementary Fig. 6a. 2D $GaPO_4$ sheets synthesised on a SiO$_2$/Si substrate were glued and connected on a conductive tab using silver paste. Inverse piezoelectric measurements were performed in contact mode by applying AC signal (driving excitation) to the conductive AFM tip with a frequency of 15 kHz, chosen to be far away from the cantilever resonance frequency (330 kHz). The forces applied to the supported and free-standing $GaPO_4$ were approximately 150 and 8 nN, respectively. The AC amplitude was swept from 0 to 4 V (0 V DC bias) while the tip was anchored at the piezo-active area chosen from the topography image. The responsive out of plane piezoelectric vibration causes the displacement of the cantilever. A lock-in amplifier was utilised to measure the resulting vertical deflection of the cantilever, which was reflected in the final output as amplitude and phase change during PFM imaging. Piezoelectric coefficient, $d_{33}$ is used to represent the quantitative vertical piezoelectric displacement produced by an out-of-plane electric field. During actual PFM experiments, the piezoresponse coefficient is addressed as an effective piezoelectric constant, $d^{eff}_{33}$, due to possible factors that affect the PFM measurements such as inhomogeneous electric field and other electrostatic effects[10,64]. The out of plane piezoresponse amplitude variation was calculated from statistical distribution of the piezoresponse amplitude values of $GaPO_4$ film and the substrate/background (Fig. 4b–f, Supplementary Figs. 8, 12), which can be determined by:

Piezoresponse amplitude variation, $V_{PFM}$ (mV) = normal fitting mean ($GaPO_4$ nanosheet on the substrate − normal fitting mean (substrate/background))

Error bar = normal fitting standard deviation ($GaPO_4$ nanosheet on the substrate) + normal fitting standard deviation (substrate/background).

Next the $d^{eff}_{33}$ is determined by the following calculation:

We assume an AC driving voltage was applied with amplitude $V_{in}$ (V) to the flake and the resulting piezoresponse displacement is $A_{PFM}$ (pm) which is the product of the detected vertical deflection voltage $V_{PFM}$ (mV) and deflection sensitivity of the cantilever $\delta$ (nm/V)[46,48]. Therefore, the deflection sensitivity was determined each time a cantilever is mounted or remounted. Finally, $d^{eff}_{33}$ for an applied AC driving voltage can be determined by

$$d^{eff}_{33} = A_{PFM}(pm)/V_{in}(V) \qquad (3)$$

$$A_{PFM}(pm) = [V_{PFM}(mV) \times \delta(nm/V)]/16 \qquad (4)$$

Here, a hardware gain of ×16 was used to enhance the piezoresponse amplitude signal during PFM scanning. To assure the reliability and accuracy of our measurements, PFM amplitude and phase of periodically polled LiNbO$_3$ was measured by the same PFM instrument and technique as a standard reference (Supplementary Fig. 6b).

**DFT calculations**. Hybrid DFT calculations were performed using Gaussian basis set ab initio Package CRYSTAL14[65,66]. The B3LYP hybrid exchange-correlation functional[67] was used to calculate the slab energies. For all atoms a Triple Zeta Valance basis set, with polarisation functions, was used to model the electrons[68].

**Peak-force quantitative nano-mechanical mapping**. All measurements were performed using the Peak Force QNM mode with a Dimension Icon AFM from Bruker. AFM tip "ScanAsyst—Air" was purchased from Bruker AFM probes with a force constant of 0.4 N/m, resonant frequency around 70 kHz and nominal tip radius around 2 nm. AFM tip "OTESPA-R3" with a force constant of 28 N/m (nominal tip radius around 7 nm) was used to measure the threshold break point of the suspended flake. The deflection sensitivity of the cantilever was measured on a clean sapphire sample for the calibration of the probe. After calibrating the probe,

the surface topography and nano-mechanical mapping of the free-standing GaPO$_4$ flake were collected at a constant peak force with a resolution of 512 pixels. The peak force set point was set to a force of 800 pN.

**Characterisation**. The surface thickness and topography images were collected using a Bruker Dimension Icon AFM using "Scanasyst-air" AFM tips. Gwyddion 2.36 software was employed for AFM image processing and analysis. TEM imaging and EELS analysis were obtained by JEOL 2100F TEM/STEM operating with 100 keV acceleration. ImageJ 1.50i and Gatan microscopy suite 1.8.4. softwares were used for TEM image processing. The TEM samples were prepared by direct deposition of the gallium oxide skin onto TEM grids and subsequent phosphatisation.

XRD was conducted using Bruker D8 micro-diffractometer equipped with a Vantec 500 detector and 0.5 mm collimator. X-ray was generated using a 40 kV copper source and a current of 40 mA. Raman spectra of the synthesised Ga$_2$O$_3$ and GaPO$_4$ were carried out using a Horiba Scientific LabRAM HR evolution Raman spectrometer equipped with a 532 nm laser source and 1800 mm$^{-1}$ grating. Two periods of accumulation, each with duration of 240 s, were utilised to scan each of the deposited samples. Surface chemical analysis was achieved using a Thermo Scientific K-alpha XPS spectrometer equipped with a monochromatic Al K$_\alpha$ source ($hv = 1486$ eV), and a concentric hemispherical analyser. The analyser was operated with pass energy of 100 eV to record the core-level spectra (Ga 3$d$, phosphorous 2$p$, O 1$s$, C 1$s$, etc.) and 50 eV to record the valance band edge. To remove the possibility of surface charging of the 2D GaPO$_4$ affecting the XPS data, a low-energy electron flood gun was used to add electrons of 3–5 eV to the surface. XPS data acquisition and peak fitting analysis was done using the dedicated XPS Avantage software. FTIR spectroscopic measurements were conducted on a Perkin-Elmer FTIR spectrometer with a resolution of 4 cm$^{-1}$. FEI Quanta 200 environmental scanning electron microscope (environmental scanning electron microscopy (2002) equipped with an Oxford X-MaxN 20 energy dispersive X-ray spectroscopy detector was employed for exploring the morphology of the free-standing nanosheets and blank cavities. Patterning and etching of square cavities on the SiO$_2$ substrates were performed using an FEI Scios Dual Beam FIB/SEM with a gallium beam of 7 nA.

## Data availability

The data that support the findings of this study are available from the corresponding authors upon reasonable request.

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

## Acknowledgements

We thank the facilities and technical assistance of the RMIT Micro Nano Research Facility (MNRF) and the assistance of RMIT Microscopy and Microanalysis Facility (RMMF). The authors would also like to thank the Australian Research Council, Centre of Excellence for Future Low-Energy Electronics Technologies (FLEET—CE170100039) for financial support. This work was also supported by computational resources provided by the Australian Government through the National Computational Infrastructure National Facility and the Pawsey Supercomputer Centre.

## Author contributions

The project was conceived, designed and directed by K.K.-Z., T.D. and N.S. designed the synthesis methodology and the experiments. N.S. synthesised the material and conducted AFM, SEM, PFM characterisations and PF-QNM experiments. A.Z. conducted all TEM/ SAED imaging as well as EELS and assisted in PF-QNM. N.P. oxidised all silicon wafers to produce SiO₂ wafers. M.M. contributed to all the schematic illustrations. B.J.C. carried out the fabrication of microcavities using FIB/SEM. F.H. conducted the FTIR. B.Z. performed the XRD measurements. C.F.M., R.S.D. and M.M. performed the XPS measurements. N.S., J.Z.O. and K.A.M. performed Raman spectroscopy. A.J. assisted in AFM. C.X. assisted in the PFM and PF-QNM measurements and provided helpful suggestions for PFM analysis. S.P.R. carried out all DFT calculations. K.K.-Z., T.D. and N.S. analysed all results. K.K.-Z. and N.S. prepared the manuscript. All authors revised the manuscript and provided helpful comments.

## Additional information

**Competing interests:** The authors declare no competing interests.

