## [Peer Review File · Nature Communications]

Reviewers' comments:

Reviewer #1 (Remarks to the Author):

The authors report a very interesting study of 2D GaPO₄ through a novel approach of utilizing the native oxide on liquid metal. The results are interesting. The paper in its current form, however, does not provide the sort of significant advance in conceptual understanding, novelty, or technological applicability that would be likely to excite the researchers in the community. It is recommended to be reconsidered for publication after the following comments are properly addressed. Specifically,

1. The authors mentioned that "gallium oxide skin demonstrates very strong van der Waals adhesion to different substrates". This is confusing, since vdW interaction is the weakest of the weak chemical forces. Would the interaction between Ga₂O₃ and liquid Ga be weaker than the vdW interaction between the Ga₂O₃ and the transfer substrate? This raises the question that during the transfer process, would there be a certain amount of liquid Ga attached to the transferred Ga₂O₃?

2. According to the AFM image (Fig. 2a), the authors claimed the thickness was about 1.1 nm for the majority of the synthesized 2D GaPO₄. Then the piezoresponses for GaPO₄ nanosheets with different thicknesses (two to four unit cell thick) were characterized. Were the samples with different thicknesses randomly selected or synthesized by control? If the thicknesses can be controlled, how? The authors are suggested to provide more details.

3. The authors mentioned that there is no obvious thickness-dependence for the piezoelectric constant. What is the reason? This is contrary to other known piezoelectric nanomaterials, which show dimension- or thickness-dependent piezoelectric constant. Furthermore, the authors mentioned that the 2D GaPO₄'s piezoelectric constant is approximately two times larger than that of bulk GaPO₄ crystal. What is the reasons? This seems to contradict the claim of thickness-independent piezoelectric constant.

4. The authors utilized Raman spectroscopy to prove the composition of GaPO₄ sheets. There are three strong peaks shown in Fig. 3b, and the authors cited two papers reported Raman spectra of GaPO₄ for comparison. However, in both the two papers, there is a Raman active peak appearing at about 300 cm⁻¹ at atmospheric pressure and room temperature, while this peak is absent in this paper. Meanwhile, the peak at 206.2 cm⁻¹ is different from the reported results in these two references. The authors should clarify the differences in terms of Raman active modes and label the modes in figure.

5. The authors mentioned that the piezoelectric response increased under the increasing driving bias, which was due to the strong inverse piezoelectric characteristics. This is not correct, since the amplitude of piezoelectric response cannot accurately correspond to the real piezoelectric coefficient or 'piezoelectric feature' as mentioned in the paper. Only the slope of piezoelectric amplitude vs driving bias (AC) can significantly show the piezoelectric features. Moreover, the PFM response in Fig. 5b is too weak. More reliable data are necessary for GaPO₄ flakes with different thicknesses.

6. The authors attributed the mismatch between experimental result and DFT result to non-homogeneous electric field, which is generally valid for any PFM characterization results. It is suggested that the authors should provide more details to further clarify the discrepancy between the experimental and simulation results.

7. What does 'which have been fully associated with the piezoresponse' mean in this paper? In

addition, the authors also claimed their result 'outperforms' many previous results, however, there is no citation to support it.

8. At the end, the authors claim that "suggests that they are compatible with high temperature fabrication procedures used for surface processed chips and high temperature applications." More experimental evidences are required to support this claim, as the reduction in thickness can significantly reduce the Curie temperature and the stability of the material.

Reviewer #2 (Remarks to the Author):

COMMENTS TO AUTHORS

General Comments:

The work "Pringing 2D gallium phosphate out of liquid metal" from Prof. Kalantar-zadeh's group demonstrated a simple yet feasible way to fabricate GaPO₄ nanosheets with significant piezoelectricity by taking advantage of the naturally oxidized liquid metal surface. The successful preparation of the 2D GaPO₄ nanosheets is confirmed by multiple characterisation methods. The piezoelectric property and mechanical property of the obtained nanosheets are measured. Also, DTF simulation is carried out for single-layer thick nanosheets, which reached a piezoelectric coefficient comparable to their experimental measurement. The manuscript is well organized and the results are clearly presented. After further revision being made based the comments below, I recommend the work for publication in Nature Communications.

Specific Comments:

1. The lateral dimensions of the GaPO₄ nanosheets are found to vary from several millimeters to centimeter scale and as presented in the Methods section, the dimensions of the oxide sheets varied with the diameter of gallium drop. My question are: 1) Did drops with larger diameters result in larger dimensions of the obtained GaPO₄ nanosheets; 2) If so, what are experimental limitations for fabricating larger nanosheets? Please comment.

2. The results for GaPO₄ nanosheets of different thickness (unit cell) are compared (e.g., Figure 4h) while how these samples are fabricated are absent. Are the samples prepared through a single exfoliation process or repeated exfoliation with times correspond to the number of unit cell layers? Since this directly relates to the thickness control of the method, a discussion on the matter will be very informative.

3. The elastic modulus of the nanosheets is characterized in sufficient details. I am very interested another mechanical parameter of the nanosheets-the breakdown threshold of the nanosheets. It is an important parameter for the practical utilization of the piezoelectric material. Since piezoelectric constant and modulus are accessible, knowing the breakdown threshold of the nanosheets enables one to be aware of the voltage range should be controlled to avoid failures. So I suggest the authors to further provide this result if the samples as well as the equipment are still available. It can be obtained from AFM measurements by push the probe against free-standing nanosheets until breakdown. Or ideally, if sample breakdowns have already been encountered, it will be convenient to extract the result from previous tests.

4. Line 224: Fig. 6g is mis-referenced, please provide a correct figure number.

5. The following article on liquid metal processing can be cited and commented.

a) Wang Q et al., Preparations, characteristics and applications of the functional liquid metal materials, *Advanced Engineering Materials*, 1700781, 2017

Reviewer #1 (Remarks to the Author):

General: *The authors report a very interesting study of 2D GaPO₄ through a novel approach of utilizing the native oxide on liquid metal. The results are interesting. The paper in its current form, however, does not provide the sort of significant advance in conceptual understanding, novelty, or technological applicability that would be likely to excite the researchers in the community. It is recommended to be reconsidered for publication after the following comments are properly addressed. Specifically, the following points are for the authors to consider and prepare their work.*

1. The authors mentioned that “gallium oxide skin demonstrates very strong van der Waals adhesion to different substrates”. This is confusing, since vdW interaction is the weakest of the weak chemical forces. Would the interaction between Ga₂O₃ and liquid Ga be weaker than the vdW interaction between the Ga₂O₃ and the transfer substrate? This raises the question that during the transfer process, would there be a certain amount of liquid Ga attached to the transferred Ga₂O₃?

Thank you for your comment. We have added the following statement to convey your valid remark across to the readers.

Manuscript: Page 5, Paragraph 2

“Synchrotron based studies of the liquid gallium interface have revealed that the electron density profile features a prominent minimum of the electron density distribution at the boundary between the liquid metal and its naturally grown surface oxide¹. This finding proves that there are no covalent bonds between the oxide layer and the parent metal. Furthermore, liquid gallium metal is a monatomic liquid which is by default non-polar, limiting the possibility of interaction further. Hence, interaction between the liquid metal and the surface oxide are expected to be minimal. The absence of a solid crystal structure impedes cumulative atomic interactions of liquid metal over large areas rendering any weak interactions that may occur to be localised, inhibiting macroscopic attachment². The absence of covalent bonds

between the liquid metal together with the liquid state of the parent metal hence lead to minimal interactions between the oxide and the liquid metal. The van der Waals interactions between the surface oxide and the transfer substrate, on the other hand, comprise of more robust forces between permanent dipoles. The presence of a crystalline lattice within the oxide as well as the substrate ensure that interactions may occur over larger areas, leading to macroscopic attachment^{3,4}. The high surface tension of liquid gallium further ensures that the vast majority of the liquid metal separates from the oxide during the exfoliation process. Thus minimal metal inclusions are found in the large exfoliated which are furthermore consistent with our previous reports for other metal systems⁵.”

2. According to the AFM image (Fig. 2a), the authors claimed the thickness was about 1.1 nm for the majority of the synthesized 2D GaPO4. Then the piezoresponses for GaPO4 nanosheets with different thicknesses (two to four unit cell thick) were characterized. Were the samples with different thicknesses randomly selected or synthesized by control? If the thicknesses can be controlled, how? The authors are suggested to provide more details.

Thank you for this comment. During this work we utilized a liquid metal exfoliation technique that allows exfoliating the surface oxide of liquid metals. The surface oxide of the metal is formed during a self-limiting surface reaction that follows Cabrera-Mott kinetics^{1,6}. As such the thickness of the oxide is defined by the metal and temperature. We only investigated single transfer samples (i.e. we did not attempt to transfer multiple layers onto the same substrate subsequently, which may provide a pathway towards thicker layers). We observed that the majority of the sample had a defined thickness (i.e. 0.9 nm for the oxide and 1.1 nm after phosphatisation). Thicker areas may also arise due to the folding of the exfoliated sheet. These

areas were sporadically encountered and investigated when found. We decided to provide a detailed study of areas with different thicknesses and added the following to the main and Supplementary Information:

Main manuscript: Page 8, Paragraph 1

“... occasionally some GaPO₄ nanosheets of differing thicknesses were also observed during the synthesis process (more details are provided in Supplementary Note 2).”

Supplementary Note 2: Synthesis of GaPO₄ nanosheets of differing thicknesses

2D Ga₂O₃ sheets were produced using the van der Waals printing process on a liquid gallium droplet. The isolated oxide sheets were transformed to 2D GaPO₄ using a chemical vapour method. The interfacial oxide of the gallium metal has been shown to grow in a self-limiting reaction⁷ and the thickness of the transferred Ga₂O₃ nanosheets is defined by the Cabrera-Mott process^{1,6}. The thickness of the Ga₂O₃ sheets obtained through exfoliation was found to be approximately 0.8-1 nm. Occasionally different thicknesses reaching 2-3.5 nm were observed, primarily around the edges. These thicker areas arise during the separation technique of 2D Ga₂O₃ sheets from the gallium metal due to folding of the monolayer sheets.

A strategy towards the targeted synthesis of thicker samples may involve multiple subsequent exfoliation/transformation processes onto the same substrate. The thickness of the obtained 2D GaPO₄ was found to mainly depend on the original thickness of the exfoliated 2D Ga₂O₃ sheets. The thickness profiles of two different Ga₂O₃ oxide nanosheets (0.9 and 2.6 nm thick) are presented in Supplementary Fig. 5 a,c, before the phosphatisation process. The thickness of the transformed 2D GaPO₄ was consistently 1.3 times larger than that of the original Ga₂O₃

nanosheets (Supplementary Fig. 5b,d). The observed change in thickness arises due to the recrystallisation process and the incorporation of phosphate ions into the material.

Supplementary Figure 5. AFM image of 2D Ga₂O₃ sheets imaged before (a, c) and after (b, d) conversion into 2D GaPO₄.

3. The authors mentioned that there is no obvious thickness-dependence for the piezoelectric constant. What is the reason? This is contrary to other known piezoelectric nanomaterials, which show dimension- or thickness-dependent piezoelectric constant.

Furthermore, the authors mentioned that the 2D GaPO₄'s piezoelectric constant is approximately two times larger than that of bulk GaPO₄ crystal. What is the reasons? This seems to contradict the claim of thickness-independent piezoelectric constant.

Answer to the first part of Q3: Thank you for this important comment. The thickness dependence of other piezoelectric materials such as SnSe, MoS₂ etc ^{8,9} arises due to the fact

that these materials are not intrinsically piezoelectric in the bulk, but gain piezoelectric properties when the centro-symmetry of the crystal is broken as the material's thickness approaches the monolayer-thickness. GaPO₄ is an intrinsically piezoelectric material when present as bulk or ultrathin morphologies. The primary crystal structure of GaPO₄ along z axis, at any thickness, lacks centrosymmetry, providing the basic scope for generating out-of-plane piezoelectricity (Supplementary Fig. 2c). Hence, the out-of plane piezoelectricity in GaPO₄ sheets always prevails for the synthesized 2D films, which is in obvious contrast to other 2D materials of TMDC family with clear odd–even layer dependency^{8,10}. We revisited the measurements for one unit cell thick GaPO₄ and conducted additional DFT calculations for thicker (two unit cell thick) GaPO₄ which show that there is indeed an out of plane piezoelectric property for the thicker sheets.

The following text was added to the main manuscript and we also modified the Supplementary Figure 2:

Main manuscript: Page 13, Paragraph 2

“.....of different thicknesses having repeated non-centrosymmetric structure (Supplementary Fig. 2c), leading to out-of plane piezoelectricity for the synthesised 2D sheets. The obtained experimental piezoelectric constant for different GaPO₄ sheets was found nearly within 7 to 8 pm/V (Supplementary Fig. 9) exhibiting no significant thickness dependency which is in obvious contrast to other 2D materials of TMDC family with odd–even layer piezoelectric dependency^{8,11}”

Supplementary Figure 2. **a** 1 unit cell of GaPO₄ along z axis. **b** A periodic 2×1 unit cell of GaPO₄ with a (100) plane cut from the trigonal bulk phase. This shows the top and bottom oxygen atoms in the slab are hydrogen terminated. **c** Crystal structure of few unit cell thick GaPO₄ along z axis, showing the non-centrosymmetric structure is present for mono to several-unit cell thick GaPO₄.

Answer to the second part of Q3: In our work, the vertical piezoresponse was explored for continuous and homogeneous GaPO₄ nanosheets featuring different thicknesses in the order of a few nanometer (one to four unit cells thick). Due to the very thin nature of these sheets, surface effects and termination of the crystal structure are expected to play a significant role. Bulk materials on the other hand, may feature a very slightly different structural property, leading to the modulation of the electronic properties for 2D films including changes to the vertical piezoelectric constant.

The following was added to the main manuscript:

Main manuscript: Page 13, Paragraph 1

“...The improvement in the value of d_{33}^{eff} can be possibly due to the slightly different structural properties of the ultra-thin 2D GaPO₄ films, which is several order thinner

compared to the bulk. By comparing the Raman spectrum of our synthesised 2D GaPO₄ with the bulk GaPO₄, it is seen that there are significant downshifts of both the ~420.5 and ~456.6 cm⁻¹ peaks, indicating the weakening of optical phonon vibration modes (A₁) of the PO₄ tetrahedra along z direction^{12,13}. Since the piezoelectric response of the synthesised few nanometer thick GaPO₄ arises mainly from out of plane displacements, the reduction of the stability in the lattice of GaPO₄ along z axis facilitates the piezoelectric displacement upon the applied electric field. Hence, the out of plane piezoelectric constant is larger in comparison to the bulk crystal. The finding is consistent with reports on zinc oxide (ZnO) and cadmium sulphide (CdS), where the vertical piezoelectric coefficient is enhanced when the bulk crystal is thinned down to 2D films^{14,15}.”

4. The authors utilized Raman spectroscopy to prove the composition of GaPO₄ sheets. There are three strong peaks shown in Fig. 3b, and the authors cited two papers reported Raman spectra of GaPO₄ for comparison. However, in both the two papers, there is a Raman active peak appearing at about 300 cm⁻¹ at atmospheric pressure and room temperature, while this peak is absent in this paper. Meanwhile, the peak at 206.2 cm⁻¹ is different from the reported results in these two references. The authors should clarify the differences in terms of Raman active modes and label the modes in figure.

Thank you for your important remark. The following was added to the body of the main manuscript and we also updated the figure (Fig. 3) as per your invaluable suggestions.

Main manuscript: Page 9, Paragraph 2

“.....The Raman peaks at ~420.5 and ~456.6 cm⁻¹ can be assigned to vibrations involving internal bending of the PO₄ tetrahedra, which can be denoted as A₁ vibration modes¹⁶. On the other hand, the 300 cm⁻¹ Raman peak that exists in the bulk system, which is considered to be a bending mode along the z axis^{12,16}. Therefore, it can be said that the 300 cm⁻¹ peak is absent

in the 2D sample due to geometrical confinement. The Raman peak at 206.2 cm^{-1} can be ascribed to the decoupled bending mode (E-TO5) of the PO_4 tetrahedra along the x-y axis¹². Compared with the bulk GaPO_4 , this peak is more prominent due to the geometrical confinement along z axis in the 2D sample. ”

Fig. 3 Material characterisations and electronic band properties of 2D GaPO₄. **a** XRD and **b** Raman spectra of the synthesised GaPO₄ (thick nanosheet). XPS results of **c** Ga 3d and **d** P 2p regions of the synthesised GaPO₄. **e** Enlarged view of EELS for the estimation of the fundamental bandgap and the extended EELS spectrum (inset). **f** XPS valence band analysis of GaPO₄.

5. The authors mentioned that the piezoelectric response increased under the increasing driving bias, which was due to the strong inverse piezoelectric characteristics. This is not correct, since the amplitude of piezoelectric response cannot accurately correspond to the real piezoelectric coefficient or 'piezoelectric feature' as mentioned in the paper. Only the slope of piezoelectric amplitude vs driving bias (AC) can significantly show the piezoelectric features.

Moreover, the PFM response in Fig. 5b is too weak. More reliable data are necessary for GaPO₄ flakes with different thicknesses.

Thank you for your comment. We agree with your statement and edited the main text as follows:

Main manuscript: Page 13, Paragraph 1

“...The piezoresponse displacement was found to increase steadily with higher driving bias.”

Answer to the second part of Q5: *Thank you for your valuable comments. We agree with you that the PFM response in Fig. 5b is too weak as it is showing the piezoresponse amplitude for the GaPO₄ sheet for AC driving voltage 0V.*

In order to improve the reliability of our experimental results, we conducted theoretical simulation (DFT calculation) to calculate the d_{33} for thicker (two unit cell thick) GaPO₄ sheet for an applied field of 3.1×10^7 V/cm, which corresponds to an applied voltage of 3.1 V. Previously the DFT calculation of a unit cell thick GaPO₄ was computed for an applied field of 1.1×10^7 V/cm (corresponds to an applied voltage of 1.1 V). In order to keep consistency between the applied fields for both nanosheets, we recalculated the d_{33} for one unit cell thick GaPO₄ for an applied field of 2.6×10^7 V/cm which is similar to the field strength used for the thicker sheet.

DFT calculations of 2D GaPO₄ with higher thicknesses were not calculated due to significant computational complexity. We added the following comment to the main and also updated the figure (Figure 4) as per your valuable suggestions.

Main manuscript: Page 14, Paragraph 2

“....To facilitate the calculation of unit cell polarisation, a periodic 2×1 supercell of GaPO₄ with a (100) plane was cut from the trigonal bulk phase and the top and bottom oxygen atoms in the slab were hydrogen terminated (Supplementary Fig. 2b). Single and two unit cell thick slabs were constructed from the 2×1 supercell. The thickness of the one unit cell thick slab was ~1.1 nm, corresponding to the experimentally determined sheet thickness, and ~2.2 nm for the two unit cell thick sheet.”

Main manuscript: Page 15, Paragraph 2

“For the unit cell thick slab an applied electric field of 2.6×10^7 V/cm was used (corresponds to an applied voltage of ~ 2.6 V), while a slight higher field strength of 3×10^7 V/cm was used for the bilayer slab. We calculated a theoretical value of the piezoelectric constant, d_{33} to be nearly 8.5 pm/V and 6.8 pm/V in magnitude, respectively, for the single and two unit cell thick slabs indicating the slab undergoing positive strain in response to an increasing field, which is consistent with experiments. The value of experimental d_{33}^{eff} for the single unit cell thick GaPO₄ sheet is nearly in good agreement with DFT calculations with a slight mismatch. On the other hand, the experimental value of d_{33}^{eff} (~ 6.8±0.5 pm/V) of two unit cell thick GaPO₄ sheet that is in good agreement with the coefficient obtained from the DFT computational results. DFT calculations show that there is indeed out of plane piezoelectric property for the thicker (two unit cell thick) GaPO₄ sheet.

*In order to improve the accuracy of the experimentally determined piezoelectric constant further, statistical distributions for piezoresponse amplitude of the entire GaPO₄ flakes and the substrate (background) were used instead of a linescan. Error bars were also introduced to indicate uncertainty of the measurements. We also modified and edited the calculation procedure of the piezoresponse amplitude changes to further improve the reliability of the piezoelectric constants following a recently published method¹⁷. We calculated the values of d_{33}^{eff} for different GaPO₄ sheets using the measured statistical distributions and included additional discussions to in the **methods section** and added new **Supplementary Figure 8**. We also updated Figure 4 (main manuscript) and Supplementary Figure 9 accordingly to reflect the modified piezoelectric constants.*

We added the following comments:

Main manuscript: Page 13, Paragraph 1

“...The insets represent the statistical distributions of piezoresponse amplitude variations of the GaPO₄ sheet and the substrate. The statistical distributions of the piezoresponse amplitude variations provide the opportunity to characterise the overall effective piezoelectric constant (d_{33}^{eff}) for the entire flake area¹⁷.”

Methods : Page 23, Paragraph 1

“The out of plane piezoresponse amplitude variation was calculated from statistical distribution of the piezoresponse amplitude values of GaPO₄ film and the substrate (Fig. 4 and Supplementary Fig. 8) which can be determined by:

Piezoresponse amplitude variation, V_{PFM} (mV) = Normal fitting mean (GaPO₄ nanosheet on the substrate) – Normal fitting mean (substrate)

Error bar = Normal fitting standard deviation (GaPO_4 nanosheet on the substrate) + Normal fitting standard deviation (substrate).”

Fig. 4 Characterisations of out of plane piezoelectricity of 2D GaPO_4 a AFM topography of a unit cell thick GaPO_4 nanosheet. b-f Vertical piezoresponse amplitude profiles at different AC driving voltage. The insets represent the statistical distribution of the piezoresponse amplitude values of GaPO_4 film (white colour) and the substrate (grey colour) g Average piezoresponse amplitude as a function of applied AC voltage extracted from the statistical distributions of the amplitude variations of GaPO_4 and the substrate. Error bars signify the standard deviations. h Effective piezoelectric constant for different 2D GaPO_4 (one to four unit cell thick along c axis). Error bars are indicating the deviations of slope of piezoelectric amplitude versus driving bias (AC), during the calculation of experimental d_{33}^{eff} . DFT simulations of 2D GaPO_4 nanosheets with higher thicknesses were not reported due to difficulties with accuracy and energy

convergence.”

Supplementary Figures:

Supplementary Figure 8. Statistical distribution of the piezoresponse amplitude between the GaPO₄ nanosheet and the substrate.

Supplementary Figure 9. a-c Topography and vertical piezoresponse amplitude profiles for different 2D GaPO₄ nanosheets (two to four unit cell thick) at various AC driving voltage. **d-f** Average piezoresponse amplitude as a function of applied AC voltage for different GaPO₄ nanosheets obtained from the statistical distributions. Error bars denote the standard deviations.

6. The authors attributed the mismatch between experimental result and DFT result to non-homogeneous electric field, which is generally valid for any PFM characterization results. It is suggested that the authors should provide more details to further clarify the discrepancy between the experimental and simulation results.

It is assumed that the experimental results represent the real piezoelectric behavior of the nanosheet under the applied electric field with specific measuring errors (instrumental errors). The simulation results, on the other hand, represent the piezoelectric behavior of the nanosheet based on a theoretical and mathematical model that has been developed for determining the desired parameters. During our DFT simulation the developed unit cell thick GaPO₄ was assumed to be a perfectly flat sheet without any defect. Additionally, the calculations are performed considering a completely homogeneous applied electric field normal to the surface of the entire flake area, which is in obvious contrast with the real piezoelectric measurement of the nanosheet. The discrepancy between the real performance and the simulation result is possibly due to the difference between the real 2D sheet and its assumed model (either physical or mathematical). For instance, previous simulation measurements published on the piezoelectric properties of gallium nitride (GaN) showed consistency with the experimental results to within ~20%¹⁸. Our calculated values represent the variation within the accuracy, highlighting that the developed model (including all assumptions and simplifications) is overall valid.

We added the following comment to the main

Main manuscript: Page 15, Paragraph 3

“The discrepancy between the real piezoelectric performance and the simulation result is possibly due to the difference between the real 2D sheet and its assumed model (either physical or mathematical). During our DFT simulation the developed unit cell thick GaPO₄ was assumed to be a perfectly flat sheet without any defect. Additionally, the DFT calculations are performed considering a completely homogeneous applied electric field normal to the surface of the entire flake area, which is in obvious contrast with the real piezoelectric measurement of the nanosheets¹⁹.”

7. What does ‘which have been fully associated with the piezoresponse’ mean in this paper? In addition, the authors also claimed their result ‘outperforms’ many previous results, however, there is no citation to support it.

Answer to the first part of Q7:

In our paper by mentioning ‘which have been fully associated with the piezoresponse’, we wanted to indicate that the out-of-plane displacement for our synthesised 2D GaPO₄ is due to the piezoelectric property without any significant substrate effect. Hence, no obvious amplitude contrast is evident at high voltages between the free-standing GaPO₄ and the supported film.

Some previous report on the experimental value of d_{33} for 2D materials, have shown that the experimental piezoelectric effect was possibly due to the substrate effect or flexoelectric effect. Zelisko M. et al., showed a possible out-of-plane polarisation component for graphene nitride (C₃N₄) due to lattice mismatch between the flake and the substrate leading to an induced

strain²⁰. da Cunha Rodrigues et al., reported a vertical piezoresponse of single-layer graphene layers which can be attributed to the chemical interaction of graphene atoms with underlying oxygen from SiO₂ substrate²¹. Brennan Christopher J. et al., also showed that monolayer MoS₂ has a measurable out-of-plane electromechanical response originated from the flexoelectric effect rather than the piezoelectric effect²².

Answer to the second part of Q7:

Thanks for this comment. We have included relevant references as suggested.

8. At the end, the authors claim that “suggests that they are compatible with high temperature fabrication procedures used for surface processed chips and high temperature applications.” More experimental evidences are required to support this claim, as the reduction in thickness can significantly reduce the Curie temperature and the stability of the material.

Thank you for your valid and important comment. We conducted a number of experiments to present a clear answer to this comment and to explore the thermal stability of the synthesised GaPO₄ flakes, by annealing them at high temperatures. It was observed that the synthesized 2D GaPO₄ is stable at 600 °C. Degradation of the GaPO₄ nanosheets was found to begin when the samples were annealed at 700 °C.

We added the following comment to the main. We also added texts and figures to the Supporting Information and modified Supplementary Table 1 accordingly:

Main manuscript: Page 6, Paragraph 2

“...Furthermore, it was observed that the synthesized 2D GaPO₄ is stable up to 600°C (more details regarding the *experiments* to explore the thermal stability of the synthesized GaPO₄ flakes are presented in Supplementary Note 3). Degradation of the GaPO₄ nanosheets was found

to begin when the samples were annealed at 700°C.”

Conclusion: Page 20

“Furthermore, these films are compatible with high temperature fabrication procedures used for surface processed chips and high temperature applications up to 600°C.”

Supplementary Note 3: Thermal stability of the synthesized 2D GaPO₄

The thermal stability of the synthesised GaPO₄ nanosheets was investigated *via* XPS of annealed samples. For samples annealed at 600°C, the characteristic main broad peak for the Ga 3*d* region is centered at 21.4 eV and the main broad peak in the phosphorus 2*p* region is centered at a binding energy of 134.7 eV (Supplementary Fig 13a-b), which is consistent with unannealed GaPO₄ (Fig. 3c), evidencing thermal stability at this temperature. When annealed at 700°C (Supplementary Fig 13c-d), the XPS results show a significant shift of the main Ga 3*d* peaks to a lower binding energy (20.6 eV) which resembles the spectrum of Ga₂O₃²³. Furthermore, several new P 2*p* peaks appear at lower binding energies ranging from 126 to 132 eV, indicating the partial transformation from phosphate to other phosphorous compounds^{24,25}. Raman spectrum of GaPO₄ nanosheet on SiO₂ substrate annealed at 600°C shows two strong Raman peaks at ~420.8 and ~456.3 cm⁻¹ (Supplementary Fig 14) that are in good agreement with the peak positions of the original GaPO₄ nanosheet (Fig. 3b), validating the stability of the 2D GaPO₄ at temperatures up to 600°C.

Supplementary Figure 13. XPS results of **a** Ga 3d and **b** phosphorus 2p regions of the synthesised GaPO₄ annealed at 600°C. XPS results of **c** Ga 3d and **d** phosphorus 2p regions of the synthesised GaPO₄ annealed at 700°C.

Supplementary Figure 14. Raman spectrum of GaPO₄ nanosheet on SiO₂ substrate annealed at 600°C.

Supplementary Table 1. Comparison of d_{33} between some previously reported 2D films and this work.

Material	Thickness (nm)	Theoretical simulation d_{33} (pm/V)	Experimental effective d_{33} (pm/V)	Temperature stability (°C)	Ref.
ZnO	~2	-	23.7	< 350	14
CdS	2-3	-	16.4 ^a	< 600	15
Graphene on SiO ₂	0.34	1,400 ^b		-	21
MoSSe	1.4	5.24	-	< 550	26
MoSeTe	1.4	6.21	-	< 550	26
MoSTe	1.4	10.57	-	< 550	26
WSSe	1.4	5.31	-	< 550	26
WSeTe	1.4	6.71	-	< 550	26
WSTe	1.4	9.27	-	< 550	26
C ₃ N ₄	~2	0	1	-	20
MoS ₂	0.7	-	1.35±0.24 (flexoelectric effect)	< 550	22
α-GaPO ₄	~1.1	~8.5	7.5±0.8	< 700	This work

Reviewer #2 (Remarks to the Author):

General:

The work “Printing 2D gallium phosphate out of liquid metal” from Prof. Kalantar-zadeh’s group demonstrated a simple yet feasible way to fabricate GaPO₄ nanosheets with significant piezoelectricity by taking advantage of the naturally oxidized liquid metal surface. The successful preparation of the 2D GaPO₄ nanosheets is confirmed by multiple characterisation methods. The piezoelectric property and mechanical property of the obtained nanosheets are measured. Also, DFT simulation is carried out for single-layer thick nanosheets, which reached a piezoelectric coefficient comparable to their experimental measurement. The manuscript is well organized and the results are clearly presented. After further revision being made based the comments below, I recommend the work for publication in Nature Communications. Specific Comments:

The following points are for the authors to consider and prepare their work.

- 1. The lateral dimensions of the GaPO₄ nanosheets are found to vary from several millimeters to centimeter scale and as presented in the Methods section, the dimensions of the oxide sheets varied with the diameter of gallium drop. My question are: 1) Did drops with larger diameters result in larger dimensions of the obtained GaPO₄ nanosheets;*
- 2) If so, what are experimental limitations for fabricating larger nanosheets? Please comment.*

Thank you for your valuable remark. We added the following texts and image in the Supporting Information (Supplementary Note 1 and Supplementary Fig. 1).

Supplementary Note 1: Synthesis of large Ga₂O₃ nanosheets

The naturally occurring 2D layer of Ga₂O₃ fully covers the surface of the droplet in oxygen containing environment. The oxide layer is transferred by firmly touching the target substrate against the gallium droplet as reported to our previous published work⁵. Different gallium droplets with varying diameters of 4 to 20 mm were chosen (Supplementary Fig. 1a-b) to harvest Ga₂O₃ nanosheets. Based on the technique used for transferring the oxide layer, the size of the gallium droplet is not a limiting factor. The Ga₂O₃ sheets with lateral dimension of several millimetres were obtained from the gallium droplets with diameter range 1-2 centimetre. These large Ga₂O₃ sheets were later transformed into GaPO₄ by using a chemical vapour method.

However, the experimental limitations for fabricating larger nanosheets mainly depend on the applied force and the approaching angle of the target substrate that is brought into contact with the liquid metal. If the larger liquid-metal droplet is not uniformly touched by the substrate, the transferred area of Ga₂O₃ sheets may lack homogeneity. Conversely, when excessive force is used on the droplet the probability of metal inclusions increases during the oxide layer transfer occurs. The metal inclusions can be cleaned and removed by following the methods proposed by Jing Liu *et al*²⁷. During the separation of the 2D nanosheets from the gallium metal, the possibility of overlapping and folding of the monolayer along the edges result in increased flake thickness, which can also be considered as one of the limitations to this process (Supplementary Fig. 3a).

Supplementary Figure 1 . a Fresh liquid gallium droplets with different diameters before the formation of Ga_2O_3 oxide skin. **b** Gallium droplets with surface oxide skin.

2. *The results for GaPO_4 nanosheets of different thickness (unit cell) are compared (e.g., Figure 4h) while how these samples are fabricated are absent. Are the samples prepared through a single exfoliation process or repeated exfoliation with times correspond to the number of unit cell layers? Since this directly relates to the thickness control of the method, a discussion on the matter will be very informative.*

Thank you for this comment. During this work we utilised a liquid metal exfoliation technique that allows the exfoliation of the surface oxide of liquid metals. The surface oxide of the metal is formed during a self-limiting surface reaction that follows Cabrera-Mott kinetics⁶. As such the thickness of the oxide is defined by the metal and the temperature. We only investigated single transfer samples (i.e. we did not attempt to transfer multiple layers onto the same substrate subsequently, which may provide a pathway towards thicker layers). We observed that the majority of the sample had a defined thickness (i.e. 0.9 nm for the oxide and 1.1 nm after phosphatisation). Thicker areas arise due to folding of the exfoliated sheet. These areas were sporadically encountered and investigated when found. We decided to provide a detailed study of areas with different thicknesses and added the following to the SI:

Main manuscript: Page 8, Paragraph 1

“...occasionally some GaPO₄ nanosheets of differing thicknesses were also observed during the synthesis process (more details are provided in Supplementary Note 2).”

Supplementary Note 2: Synthesis of GaPO₄ nanosheets of differing thicknesses

2D Ga₂O₃ sheets were produced using the van der Waals printing process on a liquid gallium droplet. The isolated oxide sheets were transformed to 2D GaPO₄ using a chemical vapour method. The interfacial oxide of the gallium metal has been shown to grow in a self-limiting reaction⁷ and the thickness of the transferred Ga₂O₃ nanosheets is defined by the Cabrera-Mott process^{1,6}. The thickness of the Ga₂O₃ sheets obtained through exfoliation was found to be approximately 0.8-1 nm. Occasionally different thicknesses reaching 2-3.5 nm were observed, primarily around the edges. These thicker areas arise during the separation technique of 2D Ga₂O₃ sheets from the gallium metal due to folding of the monolayer sheets.

A strategy towards the targeted synthesis of thicker samples may involve multiple subsequent exfoliation/transformation processes onto the same substrate. The thickness of the obtained 2D GaPO₄ was found to mainly depend on the original thickness of the exfoliated 2D Ga₂O₃ sheets. The thickness profiles of two different Ga₂O₃ oxide nanosheets (0.9 and 2.6 nm thick) are presented in Supplementary Fig. 5 a,c, before the phosphatisation process. The thickness of the transformed 2D GaPO₄ was consistently 1.3 times larger than that of the original Ga₂O₃ nanosheets (Supplementary Fig. 5b,d). The observed change in thickness arises due to the recrystallisation process and the incorporation of phosphate ions into the material.

Supplementary Figure 5. AFM image of 2D Ga_2O_3 sheets imaged before (a, c) and after (b, d) conversion into 2D GaPO_4 .

3. *The elastic modulus of the nanosheets is characterized in sufficient details. I am very interested another mechanical parameter of the nanosheets-the breakdown threshold of the nanosheets. It is an important parameter for the practical utilization of the piezoelectric material. Since piezoelectric constant and modulus are accessible, knowing the breakdown threshold of the nanosheets enables one to be aware of the voltage range should be controlled to avoid failures. So I suggest the authors to further provide this result if the samples as well as the equipment are still available. It can be obtained from AFM measurements by push the probe against free-standing nanosheets until breakdown. Or ideally, if sample breakdowns have already been encountered, it will be convenient to extract the result from previous tests.*

Thank you for your valid and important comment. As you suggested, we measured the breakdown threshold load of the suspended GaPO_4 membrane using peak-force quantitative nano-mechanical method, as presented in the Supplementary Figure 12. A line of discussion was also added to the main manuscript.

Main manuscript: Page 18, Paragraph 2

The peak-force quantitative nano-mechanical mapping was next used for estimating the breakdown threshold load for the free-standing GaPO₄ flake by varying the force applied to the tip. The threshold force was found at around 200 nN to break the GaPO₄ membrane (Supplementary Fig. 12).

Supplementary Figure 12. **a** AFM image of a suspended GaPO₄ nanosheet (before the breakdown occurred) on a cavity with 2 μm depth. **b** AFM image of the fractured membrane after applying 200 nN load to the tip. **c** Cross sections of the AFM topographic maps of the free standing GaPO₄ at different forces.

4. **Line 224: Fig. 6g is mis-referenced, please provide a correct figure number.**

Thank you for your valuable comment. We have incorporated your remark as follows:

“....a linear relationship with the driving voltage (Fig. 4g).”

5. **The following article on liquid metal processing can be cited and commented.**
a) Wang Q et al., Preparations, characteristics and applications of the functional liquid metal materials, *Advanced Engineering Materials*, 1700781, 2017

Thank you for your comment. We have added the reference to convey your valid remark as follows:

Main manuscript: Page 20, Paragraph 1

“Liquid metals present many potential applications²⁸”

References

- 1 Daeneke, T. *et al.* Liquid metals: fundamentals and applications in chemistry. *Chem. Soc. Rev.* **47**, 4073-4111 (2018).
- 2 Chandler, D., Weeks, J. D. & Andersen, H. C. Van der Waals picture of liquids, solids, and phase transformations. *Science* **220**, 787-794, doi:10.1126/science.220.4599.787 (1983).
- 3 Koenig, S. P., Boddeti, N. G., Dunn, M. L. & Bunch, J. S. Ultrastrong adhesion of graphene membranes. *Nat. Nanotechnol* **6**, 543–546 (2011).
- 4 DelRio, F. W. *et al.* The role of van der Waals forces in adhesion of micromachined surfaces. *Nat. Mater* **4**, 629-634, doi:10.1038/nmat1431 (2005).

- 5 Zavabeti, A. *et al.* A liquid metal reaction environment for the room-temperature synthesis of atomically thin metal oxides. *Science* **358**, 332-335 (2017).
- 6 Cabrera, N. & Mott, N. F. Theory of the oxidation of metals. *Rep. Prog. Phys* **12**, 163 (1949).
- 7 Carey, B. J. *et al.* Wafer-scale two-dimensional semiconductors from printed oxide skin of liquid metals. *Nat. Commun.* **8**, 14482 (2017).
- 8 Wu, W. *et al.* Piezoelectricity of single-atomic-layer MoS₂ for energy conversion and piezotronics. *Nature* **514**, 470-474 (2014).
- 9 Fei, R., Li, W., Li, J. & Yang, L. Giant piezoelectricity of monolayer group IV monochalcogenides: SnSe, SnS, GeSe, and GeS. *Appl. Phys. Lett.* **107**, 173104 (2015).
- 10 Zhu, H. *et al.* Observation of piezoelectricity in free-standing monolayer MoS₂. *Nat. Nanotechnol.* **10**, 151-155, doi:10.1038/nnano.2014.309
<https://www.nature.com/articles/nnano.2014.309#supplementary-information> (2014).
- 11 Zhu, H. *et al.* Observation of piezoelectricity in free-standing monolayer MoS₂. *Nat. Nanotechnol* **10**, 151–155 (2014).
- 12 Souleiman, M. *et al.* Combined experimental and theoretical Raman scattering studies of α -quartz-type FePO₄ and GaPO₄ end members and Ga_{1-x}Fe_xPO₄ solid solutions. *RSC Adv.* **3**, 22078-22086 (2013).
- 13 Angot, E. A high-temperature Raman scattering study of the phase transitions in GaPO₄ and in the AlPO₄–GaPO₄ system. *J. Phys. Condens. Matter* **18**, 4315-4327 (2006).
- 14 Wang, L. *et al.* Ultrathin piezotronic transistors with 2 nm channel lengths. *ACS Nano* **12**, 4903–4908, doi:10.1021/acsnano.8b01957 (2018).

- 15 Wang, X. *et al.* Subatomic deformation driven by vertical piezoelectricity from CdS ultrathin films. *Sci. Adv.* **2**, e1600209 (2016).
- 16 Angot, E. *et al.* A high-temperature Raman scattering study of the phase transitions in GaPO₄ and in the AlPO₄–GaPO₄ system. *J. Phys. Condens. Matter* **18**, 4315 (2006).
- 17 Wang, L. *et al.* Ultrathin Piezotronic Transistors with 2 nm Channel Lengths. *ACS Nano*, doi:10.1021/acsnano.8b01957 (2018).
- 18 Duerloo, K.-A. N., Ong, M. T. & Reed, E. J. Intrinsic piezoelectricity in two-dimensional materials. *J Phys Chem Lett.* **3**, 2871-2876 (2012).
- 19 Christman, J., Maiwa, H., Kim, S.-H., Kingon, A. & Nemanich, R. Piezoelectric measurements with atomic force microscopy. *Mater. Res. Soc. Symp. Proc.* **541**, 617 (1998).
- 20 Zelisko, M. *et al.* Anomalous piezoelectricity in two-dimensional graphene nitride nanosheets. *Nat. Commun.* **5**, 4284 (2014).
- 21 Da Cunha Rodrigues, G. *et al.* Strong piezoelectricity in single-layer graphene deposited on SiO₂ grating substrates. *Nat. Commun.* **6**, 7572 (2015).
- 22 Brennan, C. J. *et al.* Out-of-plane electromechanical response of monolayer molybdenum disulfide measured by piezoresponse force microscopy. *Nano Lett.* **17**, 5464-5471 (2017).
- 23 Zhang, W. *et al.* Liquid metal/metal oxide frameworks. *Adv. Funct. Mater.* **24**, 3799-3807, doi:

(2014).
- 24 Son, C. Y., Kwak, I. H., Lim, Y. R. & Park, J. FeP and FeP₂ nanowires for efficient electrocatalytic hydrogen evolution reaction. *Chem. comm.* **52**, 2819-2822 (2016).

- 25 Huang, S. R., Lu, X., Wang, X., Barnett, A. M. & Opila, R. L. in *Photovoltaic Specialists Conference*. 1-4 (IEEE).
- 26 Dong, L., Lou, J. & Shenoy, V. B. Large in-plane and vertical piezoelectricity in janus transition metal dichalcogenides. *ACS Nano* **11**, 8242-8248 (2017).
- 27 Ma, R., Zhou, Y. & Liu, J. Erasing and Correction of Liquid Metal Printed Electronics Made of Gallium Alloy Ink from the Substrate. *arXiv:1706.01457* (2017).
- 28 Qian, W., Yang, Y. & Jing, L. Preparations, Characteristics and Applications of the Functional Liquid Metal Materials. *Adv Eng Mater.* **20**, 1700781, doi:doi:10.1002/adem.201700781 (2018).

Reviewers' comments:

Reviewer #1 (Remarks to the Author):

The authors have put significant efforts addressing my previous comments. However, there are still some serious concerns over the PFM results and the understanding of the piezoelectric results. I have follow-up comments which need to be properly addressed before this manuscript could be considered for potential publication in Nature Communications.

1. For my previous comment 3.

The authors indicated that the lack of thickness-dependent piezoelectric constant in 2D GaPO₄ is because such material is piezoelectric for all thicknesses. This is not correct. It is well-known that piezoelectric nanomaterials such as ZnO, GaN, etc show strong dimension/thickness-dependent piezoelectric property. These materials are non-centrosymmetric for bulk or nanoscale.

Also, the authors mentioned that 2D MoS₂, SnSe etc have thickness-dependent piezoelectric constant because they are non-centrosymmetric for odd-layers and centrosymmetric for even-layers. But what about the thickness-dependent piezoelectric constant for odd-layer samples? They are all non-centrosymmetric.

Moreover, the authors confirmed that the 2D GaPO₄ shows twice larger piezoelectric constant than that of bulk GaPO₄. If this is the case, then it is difficult to understand why there is no thickness-dependence for the piezoelectric constant.

2. Also, the authors used Raman result to explain the thickness-dependent piezoelectric response, which is confusing. Back to the fundamental of any 'dipole', only the distance between two charge center with different symbols and the charge density will affect the dipole intensity. The authors should provide in-depth understanding of their data.

3. The authors admitted that their figure 5b is from the PFM data under 0V AC. If so, the piezoelectric response they got should be attributed to the background effect (substrate), which is contradictory to their response in comment 7, that is "is due to the piezoelectric property without any significant substrate effect". The authors should rule out the environmental factors in their PFM test.

Reviewer #2 (Remarks to the Author):

The authors well addressed my comments and suggestions. I would like to recommend accept the article for publication.

Reviewers' comments:

Reviewer #1 (Remarks to the Author):

The authors have put significant efforts addressing my previous comments. However, there are still some serious concerns over the PFM results and the understanding of the piezoelectric results. I have follow-up comments which to be properly addressed before this manuscript need could be considered for potential publication in Nature Communications.

Thanks for your invaluable comments. We addressed them one by one and incorporated the changes as you requested as described for each of your comments.

1. For my previous comment 3.

The authors indicated that the lack of thickness-dependent piezoelectric constant in 2D GaPO₄ is because such material is piezoelectric for all thicknesses. This is not correct. It is well-known that piezoelectric nanomaterials such as ZnO, GaN, etc show strong dimension/thickness-dependent piezoelectric property. These materials are non-centrosymmetric for bulk or nanoscale. Also, the authors mentioned that 2D MoS₂, SnSe etc have thickness-dependent piezoelectric constant because they are non-centrosymmetric for odd-layers and centrosymmetric for even-layers. But what about the thickness-dependent piezoelectric constant for odd-layer samples? They are all non-centrosymmetric. Moreover, the authors confirmed that the 2D GaPO₄ shows twice larger piezoelectric constant than that of bulk GaPO₄. If this is the case, then it is difficult to understand why there is no thickness-dependence for the piezoelectric constant.

The comment is very important, and we would like to sincerely thank the reviewer for emphasising his/her significant input. The reviewer is correct. While at small changes of thickness the d_{33}^{eff} value seems to be less dependent on the thickness of the film, however, d_{33}^{eff} shows a gradual thickness dependence for GaPO₄ films with larger thicknesses. Our

extra measurements demonstrate that as the reviewer predicted the value of d_{33}^{eff} for GaPO₄ films gradually decreases with increasing thickness, demonstrating a possible decrease towards that of the bulk value reported by previous literature. Here the values of d_{33}^{eff} were measured at 6.6 and 6.1 pm/V for the 8 and 11 nm thick films, respectively. The films with larger thicknesses were prepared from repeating van der Waals printing method multiple times. Details of the PFM measurements for the thicker film is included in the new Supplementary Figure 12. These values were also included in the main manuscript Figure 4h. The discussion part is also amended to convey the valid remark by the reviewer.

Main manuscript: Page 14, Paragraph 2

“The vertical piezoresponse was also explored for GaPO₄ nanosheets of different thicknesses (Supplementary Fig. 9). The obtained experimental piezoelectric constant was measured to be within the range from 6.9 to 7.2 pm/V for two to four unit cell thick (~2.2-4.3 nm) GaPO₄ sheets (Supplementary Fig. 9a-c, 9e-g). However, more gradual thickness dependence behaviour of d_{33}^{eff} was observed for thicker GaPO₄ films which were prepared from multiple van der Waals printing process as discussed in Supplementary Note 2. Two PFM measurements for the thicknesses of 8 and 11 nm films resulted in the d_{33}^{eff} values of 6.6 and 6.1 pm/V, respectively, demonstrating a possible decrease towards that of the bulk value reported in previous literature¹.”

Fig. 4 Characterisations of out of plane piezoelectricity of 2D GaPO₄ **a** AFM topography of a unit cell thick GaPO₄ nanosheet. **b-f** Vertical piezoresponse amplitude profiles at different AC driving voltages. The insets represent the statistical distribution of the piezoresponse amplitude variations of GaPO₄ film (white colour) and the substrate (grey colour). **g** Average piezoresponse amplitude as a function of the applied AC voltage extracted from the statistical distributions of the amplitude variations of GaPO₄ and the substrate. Error bars signify the standard deviations which are introduced to indicate uncertainty of the measurements. **h** Value of d_{33}^{eff} for GaPO₄ films with different thicknesses. The bulk value for d_{33} was extracted from refernce¹. Error bars signify the deviations of slope of piezoelectric amplitude for the driving bias voltage (AC) for experimental d_{33}^{eff} . DFT simulations of 2D GaPO₄ nanosheets of higher thicknesses were not reported due to difficulties with accuracy and energy convergence.

Supplementary Figure 9. a-d Topography and vertical piezoresponse amplitude profiles for different 2D GaPO₄ nanosheets at various AC driving voltages. e-h Average piezoresponse amplitude as a function of the applied AC voltage for different GaPO₄ nanosheets obtained from the statistical distributions. Error bars denote the standard deviations.

2. Also, the authors used Raman result to explain the thickness-dependent piezoelectric response, which is confusing. Back to the fundamental of any 'dipole', only the distance between two charge center with different symbols and the charge density will affect the dipole intensity. The authors should provide in-depth understanding of their data.

Thank you for this important comment and indeed the reviewer is correct. As per response to comment 2, we have amended the paper accordingly.

To address the reviewer's comment regarding including an in-depth discussion, we would

like to clarify that the Raman spectroscopy used here was to assess the possible structural variation of 2D GaPO₄ compared to its bulk counterpart. It is hypothesized that the enhancement of piezoelectric response in 2D GaPO₄ is likely due to a slightly disordered crystal structure exhibited by the 2D component. From the Raman spectra (Fig. 3b), it is observed that there are significant shifts to lower wavenumbers for both of the ~420.5 and ~456.6 cm⁻¹ peaks in 2D GaPO₄ with reference to their bulk values, indicating the weakening of optical phonon vibration modes (A₁) of the PO₄ tetrahedra along z direction.^{2,3} As a result, the PO₄ tetrahedra in the 2D GaPO₄ crystal structure has a relatively low stability and trends to become more disordered along the z-direction when external stimuli are applied (e.g. external voltage in this case), due to the crystal's enhanced asymmetry compared to the bulk counterpart. Therefore, the piezoelectric response resulted in the z-direction is expected to augment in the 2D form.

The following section is also included to the main paper to expand the discussion as requested:

Main manuscript: Page 10, Paragraph 1

“...Another interesting observation is that the Raman peak intensity for 456.6 cm⁻¹ peak is significantly reduced compared to that of 420.5 cm⁻¹ for 2D GaPO₄ in comparison to the bulk counterpart^{2,3}. It has been reported that the A₁ Raman mode is sensitive to the free charge carrier density in graphene and 2D metal chalcogenides⁴⁻⁷. 2D GaPO₄ has a wider bandgap which is likely to include more trap states. The emergence of more polar 2D GaPO₄ modifies the interaction between phonon and free charge carriers, generated by the trap states within the 2D material, leading to phonon self-energy renormalisation. Consequently, phonons are weakened, causing the intensity of charge-sensitive A₁ Raman mode to be reduced⁴⁻⁷. Therefore, the observation from Raman spectra suggests that there an increase in the dipole

intensity in 2D GaPO₄.”

The following section is also modified to the main paper:

Main manuscript: Page 14, Paragraph 1

“ The improvement in the value of d_{33}^{eff} can be possibly due to a slightly disordered crystal structure exhibited by the 2D GaPO₄ films. From the Raman spectra (Fig. 3b), it is observed that there are significant shifts to lower wavenumbers for both of the ~ 420.5 and ~ 456.6 cm⁻¹ peaks in 2D GaPO₄ with reference to their bulk values, indicating the weakening of optical phonon vibration modes (A_1) of the PO₄ tetrahedra along the z direction.^{2,3} As a result, the PO₄ tetrahedra in the 2D GaPO₄ crystal structure has relatively low stability and trends to become more disordered along z direction when external stimuli are applied, due to the crystal’s enhanced asymmetry compared to the bulk counterpart. Therefore, the piezoelectric response resulted in the z direction is expected to augment in the 2D form.”

3. The authors admitted that their figure 5b is from the PFM data under 0V AC. If so, the piezoelectric response they got should be attributed to the background effect (substrate), which is contradictory to their response in comment 7, that is “is due to the piezoelectric property without any significant substrate effect”. The authors should rule out the environmental factors in their PFM test.

Thanks for the valuable comment. It has been discussed in main manuscript previously that a nonzero vertical piezoresponse signal is observed even at 0V around the cavity edges (Fig 5b) which is probably due to the deviation of the perfect flatness and emergence of additional strain gradients that arise due to the sharp depth profile of the cavity⁸, resulting in a possible non-zero piezoelectric polarization component⁸

The following section was added previously in the main manuscript

Main manuscript: Page 18, Paragraph 2

“Relatively high piezoresponse deflection signals around the cavity edges are observed (even at 0V). The possible origin of the non-zero signal near the cavity edges is the emergence of additional strain gradients that arise due to the sharp depth profile of the cavity⁸. Hence we focus on the flat and uniform surface area of free-standing GaPO₄ for further analysis (area A, Fig. 5a and supplementary Fig. 12) to avoid any contribution of non-zero out-of-plane polarization.”

As per response to your comment 3, the out of plane piezoelectric response for the free-standing film has been further investigated after subtracting the background value (0V). More details on the statistical distribution of the piezoresponse amplitude variations of the free standing GaPO₄ films and background are provided in the new Supplementary Figure 12. We have focused on the flat and uniform surface area of free-standing GaPO₄ (area A) to avoid any contribution of pseudo piezoresponse component due to the deviation of the perfect flatness and non-uniform thickness.

The following text was added to the main manuscript

Main manuscript: Page 19, Paragraph 1

“...(details of calculation regarding the statistical distributions of the piezoresponse amplitude variations of the free standing GaPO₄ film and background (0V) is provided in Supplementary Figure 12).”

We have also removed the following line as the sentence was badly worded.

Main manuscript: Page 16, Paragraph 3

“...there are few experimental reports of d_{33} values for 2D materials, which have been fully associated with the piezoresponse.”

We readjusted the scales for Figure. 5(a-d) and a new figure is added to Supplementary as follows:

Fig. 5 Piezoelectric characterisations and nano-mechanical property of free-standing 2D GaPO₄ nanosheet. **a** AFM topography and **b-d** vertical piezoresponse of free-standing GaPO₄ nanosheet at different AC drive excitation. **e** AFM image of a free-standing GaPO₄ flake with DMT modulus map (inset).

Reviewer #2 (Remarks to the Author):

The authors well addressed my comments and suggestions. I would like to recommend accept the article for publication.

We would like to thank the reviewer for the support in this round of review and great feedback in the first round of review.

References

- 1 Yan, H., Ning, H., Kan, Y., Wang, P. & Reece, M. J. Piezoelectric ceramics with super high curie points. *J. Am. Ceram. Soc.* **92**, 2270-2275 (2009).
- 2 Souleiman, M. *et al.* Combined experimental and theoretical Raman scattering studies of α -quartz-type FePO_4 and GaPO_4 end members and $\text{Ga}_{1-x}\text{Fe}_x\text{PO}_4$ solid solutions. *RSC Adv.* **3**, 22078-22086 (2013).
- 3 Angot, E. A high-temperature Raman scattering study of the phase transitions in GaPO_4 and in the AlPO_4 - GaPO_4 system. *J. Phys. Condens. Matter* **18**, 4315-4327 (2006).
- 4 Liu, J. *et al.* The dependence of graphene Raman D-band on carrier density. *Nano Lett.* **13**, 6170-6175 (2013).
- 5 Rakesh, V., Barun, D., Chandra Sekhar, R. & Rao, C. N. R. Effects of charge transfer interaction of graphene with electron donor and acceptor molecules examined using Raman spectroscopy and cognate techniques. *J. Phys. Condens. Matter* **20**, 472204 (2008).
- 6 Shi, Y. *et al.* Selective decoration of Au nanoparticles on monolayer MoS_2 single crystals. *Sci. Rep* **3**, 1839 (2013).
- 7 Ou, J. Z. *et al.* Physisorption-based charge transfer in two-dimensional SnS_2 for selective and reversible NO_2 gas sensing. *ACS Nano* **9**, 10313-10323 (2015).
- 8 Zelisko, M. *et al.* Anomalous piezoelectricity in two-dimensional graphene nitride nanosheets. *Nat. Commun.* **5**, 4284 (2014).

REVIEWERS' COMMENTS:

Reviewer #1 (Remarks to the Author):

The authors have addressed my comments. I recommend acceptance of this manuscript

Reviewers' comments:

Reviewer #1 (Remarks to the Author):

The authors have addressed my comments. I recommend acceptance of this manuscript.

We would like to thank the reviewer for his/her support and also for the great feedback in the first and second round of reviews.